# Single-Cell Transcriptomics Reveals the Expression of Aging- and Senescence-Associated Genes in Distinct Cancer Cell Populations

**DOI:** 10.3390/cells10113126

**Published:** 2021-11-11

**Authors:** Dominik Saul, Robyn Laura Kosinsky

**Affiliations:** 1Division of Endocrinology, Mayo Clinic, Rochester, MN 55902, USA; 2Robert and Arlene Kogod Center on Aging, Mayo Clinic, Rochester, MN 55902, USA; 3Department of Trauma, Orthopedics and Reconstructive Surgery, Georg-August-University of Goettingen, 37073 Goettingen, Germany; 4Division of Gastroenterology and Hepatology, Mayo Clinic, Rochester, MN 55902, USA

**Keywords:** cancer, aging, single-cell sequencing, gene expression, transcriptomics, next-generation sequencing, senescence

## Abstract

The human aging process is associated with molecular changes and cellular degeneration, resulting in a significant increase in cancer incidence with age. Despite their potential correlation, the relationship between cancer- and ageing-related transcriptional changes is largely unknown. In this study, we aimed to analyze aging-associated transcriptional patterns in publicly available bulk mRNA-seq and single-cell RNA-seq (scRNA-seq) datasets for chronic myelogenous leukemia (CML), colorectal cancer (CRC), hepatocellular carcinoma (HCC), lung cancer (LC), and pancreatic ductal adenocarcinoma (PDAC). Indeed, we detected that various aging/senescence-induced genes (ASIGs) were upregulated in malignant diseases compared to healthy control samples. To elucidate the importance of ASIGs during cell development, pseudotime analyses were performed, which revealed a late enrichment of distinct cancer-specific ASIG signatures. Notably, we were able to demonstrate that all cancer entities analyzed in this study comprised cell populations expressing ASIGs. While only minor correlations were detected between ASIGs and transcriptome-wide changes in PDAC, a high proportion of ASIGs was induced in CML, CRC, HCC, and LC samples. These unique cellular subpopulations could serve as a basis for future studies on the role of aging and senescence in human malignancies.

## 1. Introduction

The process of aging is associated with declining physiological functions and displays a risk factor of various diseases including cardiovascular, musculoskeletal, and neurodegenerative conditions as well as cancer [1,2,3]. While cardiovascular diseases (CVDs) reflect a major health burden, recent studies have demonstrated that cancer has replaced CVDs as the leading cause of death in various countries [4,5]. In fact, it is estimated that the cancer burden will rise by 47% in the next 20 years due to demographic changes and, consequently, the number of cancer-related deaths will significantly increase [6,7]. Therefore, researchers have aimed to dissect the hallmarks of aging to identify potential therapeutic targets to mitigate the aging process, which might result in a reduction of the incidence of aging-associated pathologies. These hallmarks include epigenetic aberrations, which can result in differential gene expression patterns, deregulated intercellular communication, senescence, stem cell exhaustion, and genomic instability [8,9].

Next-generation sequencing approaches have contributed to the transcriptome-wide decipherment of gene expression signatures’ change during aging across species and tissues [10,11]. As expected, elderly humans and mice displayed altered transcriptional profiles of genes related to cell cycle regulation, senescence, and apoptosis [12,13]. In addition, differentially expressed genes during senescence/immortalization were shown to be key regulators of interferon- and insulin growth factor-related processes, pRB/p53-associated cell cycle regulation, as well as MAP kinase, cytoskeletal, and oxidative stress pathways [14,15,16]. Certain transcriptomic changes observed in aging organisms were linked to longevity [17,18,19,20,21]; however, how these gene expression changes contribute to aging-related pathologies is not fully understood.

Similar to aging, extensive research has been performed in order to identify transcriptome-wide changes underlying malignant transformation in different tissues. As revealed using chromatin immunoprecipitation (ChIP) sequencing, the levels of acetylated lysine 27 of histone 3 (H3K27ac), a marker for active promoters and enhancers indicating gene expression levels [22], were increased in tumor samples compared to controls [23,24]. Cancer is a highly heterogeneous group of diseases, which, depending on the cancer entity and subclass, can differ in incidence rate, prognosis, microenvironment, molecular mechanisms, and therapy resistance [25,26]. Different incidence rates in the elderly suggest that aging-related gene signatures have a differential impact on the development of distinct cancer entities [27]. In addition, cellular heterogeneity might contribute to the formation of distinct tumor cell subpopulations displaying deregulated aging-associated genetic signatures [28,29]. Single-cell mRNA-seq is an emerging technique improving our understanding of the complex intratumoral heterogeneity and whether only specific cellular subpopulations display a deregulation of aging-associated genes.

Given the scarce knowledge on the relationship between cancer- and ageing-associated transcriptional changes, we sought to perform multidimensional analyses of human bulk and single-cell RNA-sequencing data of various origins. For this purpose, we investigated bulk mRNA-seq data of different solid and hematological cancers for the expression of genes upregulated during cellular aging. We then aimed to evaluate whether only distinct cellular subpopulations display these transcriptomic changes using scRNA-seq datasets of these diseases. Our findings will contribute to the understanding of aging-related expression changes present in certain cancer entities, which might serve as a guiding principle for future studies of the role of aging and senescence in human cancer.

## 2. Materials and Methods

### 2.1. Aging Signatures

In order to define aging-associated expressional patterns, we combined 17 previously established and experimentally verified aging signatures: CellAge and GenAge [30], Casella et al. 2019 [31], SENESCopedia by Jochems et al. [29], Hernandez-Segura et al. 2017 [32], Magalhaes aging up [12], Fridman senescence up [14], and Purcell et al. 2014 [33] as well as the following MSigDB gene sets (^©^ Broad Institute, Inc., Massachusetts Institute of Technology, and Regents of the University of California, Cambridge, MA, USA): Biocarta longevity pathway (M13158), GOBP cell aging (M14701), Reactome cellular senescence (M27188), Tang senescence Tp53 targets up (M11850), WP tca cycle in senescence (M40058), WP senescence and autophagy in cancer (M39619), GOBP regulation of cell aging (M16568), GOBP positive regulation of cell aging (M24705), and GOBP replicative senescence (M14683). Due to their upregulation during cellular aging and senescence, these genes are referred to as aging/senescence-induced genes (ASIGs) throughout this manuscript [8,9]. A list of these genes can be found in Appendix A. All gene lists are supplied in Appendix A.

### 2.2. Analysis of Bulk mRNA Sequencing Data

Two publicly available bulk mRNA sequencing datasets were obtained per cancer entity (CML: GSE100026, CRC: GSE50760, HCC: GSE105130, and GSE148355 [34,35], LC: GSE81089 and GSE40419 [36,37], and GSE144119 [38,39], PDAC: GSE119794, and E-MTAB-3494, [40,41], and GSE146009 [42,43,44,45]). Sample characteristics are summarized in Appendix A. In the first CML dataset (GSE100026), peripheral blood mononuclear cells (PBMCs) from CML patients in the chronic phase and five control samples were compared [38]. In the second CML dataset (GSM4280636), PBMCs from 16 CML patients in the chronic phase and 6 control samples were used [39]. Quality control of fastq files was carried out via FastQC and reads were mapped to the human reference genome GRCh38.p10 using HISAT2 (version 2.0.3.3) on Galaxy [46]. Read count files were generated using the featureCounts tool [version 1.4.6.p5] and normalized as analyzed for differential gene expression, using DESeq2 for the PDAC gene set without provided raw counts (Soren Müller) [version 2.11.40.6]. For the other gene sets, the raw counts were first converted into a matrix (DESeqDataSetFromMatrix), before DESeq2 (1.32.0) was used. The differential expression (DE) analysis was likewise performed with DESeq2 (lfcThreshold = 0, alpha = 0.1, minimum count = 0.5). Significantly differentially regulated genes were selected by a Benjamini–Hochberg-adjusted *p*-value < 0.05 and log2-fold changes ≥0.75. For pairwise dataset comparisons, we focused on upregulated genes in order to track their enrichment in cancer. The upregulated genes were selected by applying a Benjamini–Hochberg-adjusted *p*-value < 0.05 and log2-fold changes above 0.75. An exemplary RNA-seq analysis vignette was provided as an R notebook (RNA_seq_PDAC.Rmd). These criteria were used consistently, and no further ranking was used to restrict the results in order to not take advantage of single datasets. Gene set enrichment analysis (GSEA, v. 4.1.0, Broad Institute, Inc., Massachusetts Institute of Technology, and Regents of the University of California, Massachusetts, CA, USA) [47,48] was performed using default settings (1000 permutations for gene sets, Signal2Noise metric for ranking genes).

### 2.3. Single-Cell RNA-seq (scRNA-seq) Analysis

Transcriptome-wide analyses on a single cell level of human CML, CRC, HCC, LC, and PDAC were based on previously published scRNA-seq datasets [49,50,51,52,53]. Within the CML dataset, plasma cells from multiple myeloma (MM) patients were used. For the comparison of all ASIGs, three healthy bone marrow plasma cell samples (GSM5332301, GSM5332302, and GSM5332303) were included. The annotation was provided by the authors of the original study, and only plasma cells were provided (“orig.ident”) [49]. Subsequently, healthy plasma cells were used as controls for comparison purposes, and “MM” samples as malignant samples. Seurat normalization and clustering followed the recommendations of the Seurat package [54]. The multiple myeloma plasma cells were depicted in Similarity Weighted Nonnegative Embedding (SWNE), aligned along pseudotime following the monocle package, and analyzed regarding their distinct gene expression along kernel density estimation ([49] GSE175385). The annotation for the CRC dataset (“Cell_type”, “Class”) was provided by the authors in the original publication, where epithelial cells were classified as normal (class “Normal”) or malignant (class “Tumor”) [51]. Malignant cells were used for SWNE, pseudotime, and kernel density investigations. For the whole genome comparison, only control and malignant epithelial cells were compared ([51] GSE132465). The annotation for the HCC dataset was provided in the original study, where it was referred to as “Type” [52]. The HCC dataset contained malignant hepatocytes as well as tumor-associated macrophages, T cells, B cells, cancer-associated fibroblasts, tumor-associated endothelial cells, and unclassified cells. All of them (“Other”) were compared to malignant hepatocytes for the overall comparison (“Malignant cells”) ([52] GSE151530). Malignant cells were used for downstream SWNE, pseudotime, and kernel density analyses. For analysis of the LC samples, data from 44 patients were obtained and lymphatic, normal, and metastatic brain tissue were removed, resulting in 11 tumor samples. The annotation for the LC dataset was provided by the authors of the original study (“Cell_type”, “Cell_subtype”) [50]. For the DE analysis, within the “Cell_type”: Epithelial cells, we compared the “Cell_subtype”: Malignant cells to all other cells. After sorting for malignant cells using the Seurat package v4.0.3 in R4.0.3 (R Core Team (2021). R: A language and environment for statistical computing. R Foundation for Statistical Computing, Vienna, Austria.), we created a malignant epithelial cell Seurat file for the subsequent SWNE depiction, monocle analysis, and kernel density estimation ([50] GSE131907). For the overall comparison, we used malignant epithelial cells and compared these to non-malignant epithelial cells from the same dataset. For the analysis of PDAC samples, the annotation was provided earlier [53] (“cluster”). We selected ductal cell type 2 as malignant cells for subsequent steps, as suggested by the authors ([53] PRJCA001063). For comparison purposes, we used ductal cell type 1 as a benign comparator (“Control”: Ductal cell type 1”, “Malignant”: “Ductal cell type 2”). Ductal cells type 2 were used for SWNE, monocle, and kernel density estimation. The number of cells that were contributed by each donor is shown in Appendix A. For all datasets, the pseudotime trajectory was calculated following the recommendations by Trapnell [55]. In detail, a lower detection limit of 0.5 was selected, and dimensional reduction with “DDRTree”, a maximum of four components, no increase of expression values before dimensional reduction (pseudo-expression = 0), and scaling of each gene before trajectory reconstruction (scaling = TRUE). The differential gene test was performed according to the formula “~sm.ns (Pseudotime)”, with relative expression values (relative_expr = TRUE). Subsequently, just the significant genes were subset (qval < 0.1) and plotted on the heatmap. Comparisons between control and malignant cells were calculated with the “FindMarkers” function (Seurat package), while using the “MAST” package [56] (FindMarkers (ident.1 = ”Control”, ident.2 = ”Malignant”, test.use = ”MAST”, min.cells.feature = 3, min.cells.group = 3, min.pct = 0.1, logfc.threshold = 0.25). The utilized R-packages for downstream analyses were Nebulosa (3.13 [57]), Monocle (2.18.0 [58]), SWNE (0.6.2.0 [59]), MAST (1.16.0), and ggplot2 (3.3.5). An exemplary single-cell RNA-seq analysis vignette was provided as an R notebook (single_cell_RNA_Seq_MM.Rmd).

### 2.4. Statistics and Graphs

Statistical analyses were performed using a D’Agostino and Pearson test for normality. If passed, an unpaired *t*-test was performed. Otherwise, a Mann–Whitney test was performed (* *p* ≤ 0.05, ** *p* ≤ 0.01, *** *p* ≤ 0.001). Graphs were designed using GraphPad Prism 9.2.0 (GraphPad Software, Inc., San Diego, CA, USA), BioRender.com (accessed on 14 September 2021), and R (4.0.3, R Core Team (2021). R: A language and environment for statistical computing. R Foundation for Statistical Computing, Vienna, Austria.).

## 3. Results

### 3.1. Cancer Incidence Increases with Age and Bulk mRNA-seq Analyses Reveal Underlying Transcriptomic Changes

The significant increase in the incidence of malignant diseases during aging is known, and despite cancer-specific differences, the link between this group of diseases and age is apparent. The rise in cancer cases per 100,000 [60] in the young and elderly population (Figure 1A) implies a deregulation of cancer-relevant molecular pathways during aging. In this study, we focused on one hematological and four solid cancer entities, which all displayed drastic increases in the number of cases when comparing the age group below 30 years to individuals older than 80 years, i.e., in chronic myelogenous leukemia (CML; 33.6-fold), hepatocellular carcinoma (HCC; 151.5-fold), colorectal cancer (CRC; 238.4-fold), pancreatic ductal adenocarcinoma (PDAC; 443.3-fold), and lung cancer (LC; 1484.7-fold). Given the time-dependent accumulation of cancer cases, we aimed to perform transcriptome-wide profiling of cancer and aging. For this purpose, we analyzed publicly available bulk mRNA-seq data sets comparing malignant to healthy control samples for each of the aforementioned diseases. We identified significant gene expression differences and, as expected, gene set enrichment analysis of oncogenic signature genes (C6) revealed an enrichment of key driver pathways in cancer samples (Figure 1B; Appendix A). Interestingly, all datasets displayed an enrichment of polycomb group protein-associated signatures (i.e., EZH2, BMI1, and MEL18) and the majority (CML, CRC, LC, and HCC) showed an enrichment in Ribosomal Protein S14 (RPS14)- and Retinoblastoma (RB)-related pathways. The enrichment of the respective oncogenic signatures is listed in Appendix A. In addition, we compared control and tumor groups with regards to patients’ age (Appendix A). While two datasets did not provide this information (HCC GSE105130 and PDAC E-MTAB-3494), a minor age difference (4.4 years) occurred in one lung dataset (GSE81089). In the CML dataset (GSE100026), the 24.8 year difference did not lead to a substantially higher number of upregulated genes compared to the other CML dataset (94 genes in GSE100026 vs. 100 genes in GSE144119).

To strengthen our findings, we analyzed two datasets per cancer entity, which showed high homogeneity as demonstrated by the substantial overlap between enriched oncogenic pathways (Figure 1C, Appendix A).

### 3.2. Different Cancer Entities Display a Heterogeneous Expression of ASIGs

After performing quality control and verifying high similarity between the two bulk mRNA-seq data sets per cancer entity, we evaluated to what extent the genes regulated in malignant samples are upregulated during the process of aging. For this purpose, we performed literature research and obtained 1535 aging/senescence-induced genes (ASIGs) from 17 studies ([12,14,29,30,31,32,33] and MSigDB gene sets M13158, M14701, M27188, M11850, M40058, M39619, M16568, M24705, M14683 ^©^ Broad Institute, Inc., Massachusetts Institute of Technology, and Regents of the University of California, Cambridge, MA, USA), resulting in a total number of 1153 genes after removing duplicates (Appendix A). After confirming similar patterns between the two datasets per cancer type, we detected that the total number of ASIGs was different among cancer entities. While in total 213 genes were upregulated in HCC, 177 genes were upregulated in LC, 170 ASIGs were induced in CML, 115 in CRC, and only 63 ASIGs were induced in PDAC (Figure 2A). As expected, ASIGs that were found to be upregulated in malignant patient samples were related to cell cycle regulation as determined by gene ontology analysis (Figure 2B, Appendix A). Interestingly, we detected the enrichment of cancer-specific molecular pathways among ASIGs. Together, our findings demonstrate that different solid and hematological human cancers display an upregulation of ASIGs compared to healthy control samples.

### 3.3. scRNA-seq Analysis Reveals the Presence of Distinct Cancer Cell Populations Expressing ASIGs

While the upregulation of various ASIGs was confirmed in cancer, the presence of non-malignant cells, such as immune regulatory cells, could have falsified bulk mRNA-seq results. In contrast, scRNA-seq allows the separation and analysis of distinct cell populations [62]. Therefore, we analyzed publicly available scRNA-seq datasets [49,50,51,52,53] and focused on cell types that were described as putative origins of malignant transformation (MM: plasma cells [63]; CRC: epithelial cells [64]; HCC: hepatocytes [65]; LC: epithelial cells [66]; PDAC: ductal cells [67]). We first compared control and malignant samples (Appendix A, not provided for CML GSE175385), and except for the PDAC dataset (PRJCA001063), we found no significant age differences, which is why we decided to compare malignant and control cells exclusively from the same sample in the PDAC dataset (ductal cell type 1 cells = control and ductal cell type 2 = malignant).

To extend our previous bulk mRNA-seq-based approach, we determined which ASIGs were highly expressed in the malignant cell populations. Notably, while LC and CRC displayed an upregulation of the majority of ASIGs, transcriptional patterns of the other cancer entities were more heterogeneous (Appendix A). In addition, the top 20 up- and downregulated ASIGs in malignant compared to control cells were determined (Appendix A). Next, we focused on the ASIGs robustly upregulated in all bulk sequencing datasets we analyzed per cancer entity to perform an in-depth pseudotemporal characterization of ASIG expression. The number of upregulated genes in each of the two RNA-seq datasets and the determined ASIGs were presented in an SWNE plot (Figure 3, Figure 4, Figure 5, Figure 6 and Figure 7, panel A). Genes significantly changing expression along pseudotime (Figure 3, Figure 4, Figure 5, Figure 6 and Figure 7, panel B) were displayed in a heatmap (Figure 3, Figure 4, Figure 5, Figure 6 and Figure 7, panel C). Finally, only the terminally upregulated genes were depicted in a tSNE plot (Figure 3, Figure 4, Figure 5, Figure 6 and Figure 7, panel D).

Plasma cells were extracted from a scRNA-seq dataset from bone marrow cells isolated from 15 multiple myeloma (MM) patients (GSE175385 [49]) and weighted according to the expression levels of ASIGs (Figure 3A). An SWNE plot uses matrix factorization to visualize key features and their influence on single cells. Interestingly, the transcriptional heterogeneity of malignant cells is reflected by the scattered distribution of the ASIGs. While the SWNE plot indicated high transcriptional diversity of ASIGs in plasma cells, pseudotime analysis revealed an early and two late highly distinct cell fates (Figure 3B). Pseudotime infers the ordering of cells along a progression lineage, which is taken from their expressional profile related to capture time. When observing the expression patterns of significantly regulated genes during cell development, the induction of eight ASIGs (*CDKN1B*, *CDC25B*, *KDNA3*, *BAG3*, *SUN1*, *AKR1B1*, *AKT3*, *OPTN*) during late stages was demonstrated (Figure 3C). These genes were expressed by the majority of plasma cells; however, when merging the transcriptional profiles, one cellular subpopulation displaying a high abundance of these ASIGs was found (Figure 3D).

To test whether other cancer types show an enrichment of ASIGs in malignant cells as well, we utilized publicly available CRC scRNA-seq data (GSE132465 [51]) containing samples from 23 patients. ASIGs were broadly expressed among epithelial CRC cells (Figure 4A) and pseudotime analysis revealed an early and three late cell fates (Figure 4B). Interestingly, the majority of significantly regulated ASIGs were induced during cell development (Figure 3C). Pseudotime ordering led to six distinct clusters, and we focused on the gene cluster that displayed the strongest upregulation during development (*CDKN1A*, *MXD1*, *SLC30A10*, *ATF3*, *IL6R*). In fact, *IL6R* was only highly expressed by one epithelial cell subpopulation, which, notably, displayed co-expression of the other members of the cluster (Figure 4D).

HCC bulk mRNA-seq analysis revealed the upregulation of 94 ASIGs, which were further evaluated for their role in hepatocytes isolated from a HCC scRNA-seq experiment (GSE151530 [52]). Remarkably, 84 of these genes were found to undergo significant expression changes during cell development as confirmed using pseudotime alignment (Figure 5A–C). The significant late-stage upregulation of a subset of nine of these genes (*ACADVL*, *FOXO1*, *SOCS2*, *NFKBIA*, *LCAT*, *MT1F*, *UBB*, *RHOB*, *ESR1*) was verified in a tSNE embedding (Figure 5D), independently confirming a distinct cellular HCC subpopulation associated with an aging phenotype.

In an scRNA-seq experiment containing samples from 44 lung adenocarcinoma patients, all 11 lung tumor samples were selected (GSE131907 [50]), and all malignant epithelial cells were extracted and weighted according to robustly expressed ASIGs (Figure 6A). Their developmental stage was depicted with pseudotime, indicating four cellular fates along (pseudo-)temporal maturation (Figure 6B). Distinct clusters were assigned according to gene expressional patterns, which revealed that the expression of seven genes (*CAV1*, *PPARG*, *CXCL2*, *OASL, JUND*, *RRAS*, *APOL3*) was induced in late stages (Figure 6C). These seven genes were found to be dimensionally co-localized in one defined cellular subpopulation (Figure 6D).

Within 24 PDAC scRNA-seq samples (PRJCA001063 [53]), we aimed to identify aging/senescence-associated patterns. We found an overlap of eight genes between both PDAC bulk mRNA-seq studies and the ASIGs (Figure 2A). These eight genes were depicted in an SWNE plot with an overall homogenous appearance (Figure 7A) and pseudotime enrichment was detected for six genes (*IGFBP3, SLC16A3, COL10A1, PKM, S100A11,* and *GAPDH;*
Figure 7B,C). The expressional patterns led to a substantial overlap within one subpopulation of PDAC cells (Figure 7D).

## 4. Discussion

In this study, we aimed to identify aging/senescence-associated gene expression patterns enriched in next-generation sequencing datasets of five cancer entities. Besides demonstrating a substantial overlap of regulated genes and, accordingly, oncogenic signatures enriched in the two datasets we analyzed per cancer type, we detected the upregulation of ASIGs in a subpopulation of malignant cells. As expected, 1153 ASIGs tested in our study showed a heterogenous expression pattern in all control and malignant samples in both bulk and single-cell sequencing datasets. While scRNA-seq data for CRC, HCC, and LC revealed an upregulation of ASIGs in the majority of cancer cells, the proportion of ASIGs induced in control compared to malignant cells was similar in the bone marrow and pancreas.

Notably, all datasets displayed an enrichment of polycomb group protein-associated signatures (i.e., EZH2, BMI1, and MEL18). This group of proteins consists of epigenetic repressors modulating the transcriptional landscape, thereby controlling cell differentiation and tumorigenesis in various cancer types [68,69]. In addition, Ribosomal Protein S14 (RPS14)-related pathways were enriched in the majority of mRNA-seq studies (CML, CRC, LC, and HCC). This factor was demonstrated to regulate proliferation in cancer cells, for instance, by regulating the activity of the tumor-promoting factor c-MYC [70,71]. Another pathway that was enriched in most datasets (CML, CRC, LC, HCC) was associated with the well-described retinoblastoma (RB) tumor suppressor. This protein was generally described as a cell cycle regulator; however, numerous studies revealed its function in cell differentiation, survival, as well as epigenetic regulation in cancer cells [72,73,74].

Importantly, while bulk mRNA-seq revealed the enrichment of similar oncogenic pathways and genes among cancer entities, we observed major differences between ASIGs induced in malignant cells in scRNA-seq data. This might be due to the presence of non-malignant cells in cancer samples, for instance, immune regulatory cells, which “dilute” the gene expression results of transformed cells. We subsequently focused on the ASIGs enriched in each cancer entity individually and followed their expressional pattern in pseudotime. While the role of various ASIGs in cancer has been described in previous studies, their function as potential drivers of malignant progression and cellular aging in distinct cellular subtypes is largely unknown.

Interestingly, we found a heterogeneous distribution of the ASIGs in the SWNE plot of all analyzed cancer types. This finding suggests that the selected genes mirror the biological heterogeneity of malignant cell subpopulations.

As described earlier, pseudotime analysis displays a valuable tool to explore tumor heterogeneity and to detect different intratumoral cellular states based on gene expression patterns [75]. In our pseudotime graphs, we detected between two and four cellular subpopulations per cancer entity, which suggests a certain degree of tumor heterogeneity. Indeed, other studies have performed in-depth evaluation of intratumoral heterogeneity using pseudotime in the past [75,76]. The upregulation of specific ASIGs during pseudotemporal development was restricted to a narrow population in the tSNE plot. This finding suggests the presence of a distinct cellular population, which acquired the expression of these temporal-dependent genes, even in a malignant state.

Future studies analyzing highly purified cell populations combined with functional analyses could help to further describe intratumoral cell populations in the datasets used throughout this manuscript.

Plasma cells extracted from an MM scRNA-seq study contained one subpopulation inducing the expression of ASIGs (*CDKN1B*, *CDC25B*, *KDNA3*, *BAG3*, *SUN1*, *AKR1B1*, *AKT3*, *OPTN*). While CDKN1B/p27^Kip1^ loss-of-function was previously described to cause the Multiple Endocrine Neoplasia type 4 (MEN4 [77]), its role as a tumor suppressor has been demonstrated in sarcoma and lymphoma [78]. Increased *CDC25B* [79], *AKR1B1* [80], and *AKT3* expression [81] was associated with poor multiple myeloma survival, indicating a detrimental effect of certain aging-associated gene patterns. On the other hand, AKT3 is a potential therapeutic candidate as shown in vitro [82]. The enrichment of certain ASIGs along pseudotime and the accumulation of these genes in distinct cellular subpopulations as shown in a tSNE reflect the presence of an aging-associated subpopulation in MM cancer cells. These cells might represent a potential target for specific senolytic approaches, as discussed by Carpenter et al. [83].

The CRC bulk mRNA-seq datasets we analyzed displayed high homogeneity with a substantial overlap of upregulated genes and enriched oncogenic pathways. Only five ASIGs (*CDKN1A*, *MXD1*, *SLC30A10*, *ATF3*, *IL6R*) were stably induced during the pseudotemporal development of CRC epithelial cells. CDKN1A (p21), a cyclin-dependent kinase inhibitor (CDKi) and well-described inducing factor of cellular senescence, was previously connected to greater responsiveness to chemotherapy [84]. In addition, besides its involvement in senescence and aging, it was suggested to act as both a tumor suppressor and oncogene [85]. SLC30A10 inhibits colorectal cancer progression and metastasis and can be used both as a prognostic biomarker and anti-metastatic therapeutic target [86]. The role of ATF3 in colorectal cancer is yet to be determined [87], but the tumor suppressor MXD1 was deciphered to be targeted by MiR-19a/b in gastric cancer [88].

In hepatocytes extracted from a HCC scRNA-seq study, seven genes were robustly upregulated during pseudotime development, i.e., *ACADVL*, *FOXO1*, *SOCS2*, *NFKBIA*, *MT1F*, *UBB*, *RHOB*, *ESR1*, and *LCAT*. Interestingly, most of these genes were described to have a favorable role in HCC and in general, including FOXO1, which was shown to be associated with improved survival [89]. Interestingly, SOCS2 appeared to be the only beneficial factor in a seven-gene HCC predictive risk model by Chen et al. [90]. Independently, Liu et al. indicated that low expression of *FOXO1* and *SOCS2* was a poor survival outcome parameter for HCC [91], which, as suggested by our analysis, might be due to a loss of aging-related signatures. The low expression of *MT1F* has been linked to elevated tumor growth by Lu et al. [92].

In the LC scRNA-seq data we evaluated, seven ASIGs were stably induced over time (*CAV1*, *PPARG*, *CXCL2*, *JUND*, *RRAS*, *APOL3*, *OASL*). It has recently been demonstrated that CAV1 as well as PPARG are associated with a favorable outcome in lung adenocarcinoma [93,94], while the cytokine CXCL2 might be a predictor for immunotherapy response [95].

In the pancreatic cancer dataset, *IGFBP3* showed an increase during late development stages. Interestingly, *IGFBP3* has been demonstrated to be a pancreatic cancer biomarker in clinical trials and to induce muscle wasting in vitro [96,97]. Similarly, SLC16A3, a member of the SLC16 monocarboxylate transporter family that is known to play critical roles in tumorigenesis, has also recently been suggested as a biomarker by Yu et al. [98,99]. In addition, the splicing variant of *PKM,* PKM2, was associated with shorter patient survival [100]. Like *PKM*, *S100A11* showed an increase in late pseudotemporal development. The secretion of its protein, the enzyme regulator S100A11, led to increased cancer cell motility in vitro, and promoted pancreatic cancer cell proliferation in vivo [101,102]. Since a mutation in the key tumor suppressor p53 prevents GAPDH from nuclear translocation, the upregulation of GAPDH is common in a variety of tumors and closely related to pancreatic cancer. Furthermore, it was described to play a role in the enhanced sensitivity towards anti-glycolytic drugs [103,104,105].

Further studies will be necessary to validate which role cell populations expressing ASIGs have in human cancers. Whether these cells are drivers of tumor formation/progression or the result of tumor initiation/transformation remains to be elucidated. Cellular senescence has been proposed an anticancer mechanism for a long time due to the divisional arrest of apoptosis-resistant cancer cells [15,106]. Controversially, experimental findings suggested that senescent cells entail local inflammation in skin, bone, liver, and other tissues, allegedly promoting malignant transformation [15,107,108,109]. The appearance, fate, as well as the secretory phenotype of senescent cells in cancer environments suggest that senolytic therapy might serve as complementary cancer therapy as proposed by Dimri and others [106,110,111]. In addition, future research could address whether healthy cells with high expression of ASIGs show a higher tendency to undergo malignant transformation than other cells.

Taken together, we demonstrated that all cancer entities evaluated in this study comprise a cellular subpopulation expressing aging/senescence-associated genes. While there was only a minor enrichment of ASIGs upregulated in PDAC samples, a high proportion of ASIGs was induced in CML, CRC, HCC, and LC samples. In fact, as revealed by scRNA-seq, CRC and LC displayed the overall highest expression of ASIGs. The data presented here can serve as a basis for future studies on the role of aging in the development of specific tumors and be used to integrate further datasets for complementary analyses.

## 5. Limitations

While the age difference of the first LC dataset (GSE81089; 4.4 years) is unlikely to have influenced the DE results, the age difference in the first CML dataset (GSE100026) of 24.8 years is substantial and might have affected the overlap with ASIGs. However, the overlap with the ASIGs between both CML datasets was comparable (94 genes in GSE100026 vs. 100 genes in GSE144119). In the PDAC scRNA-seq dataset (PRJCA001063), the control patients were substantially younger compared to the tumor patients, which is why we decided to compare the ductal cell type 1 (control) vs. ductal cell type 2 (malignant) cells from the same samples for the single-cell analysis (Appendix A). In addition, it should be noted that next-generation sequencing analyses simplify the complex biological behavior of cancers and individual cancer cells while important interactors and their influence on the cancer cell are not taken into account.

## Figures and Tables

**Figure 1 cells-10-03126-f001:**
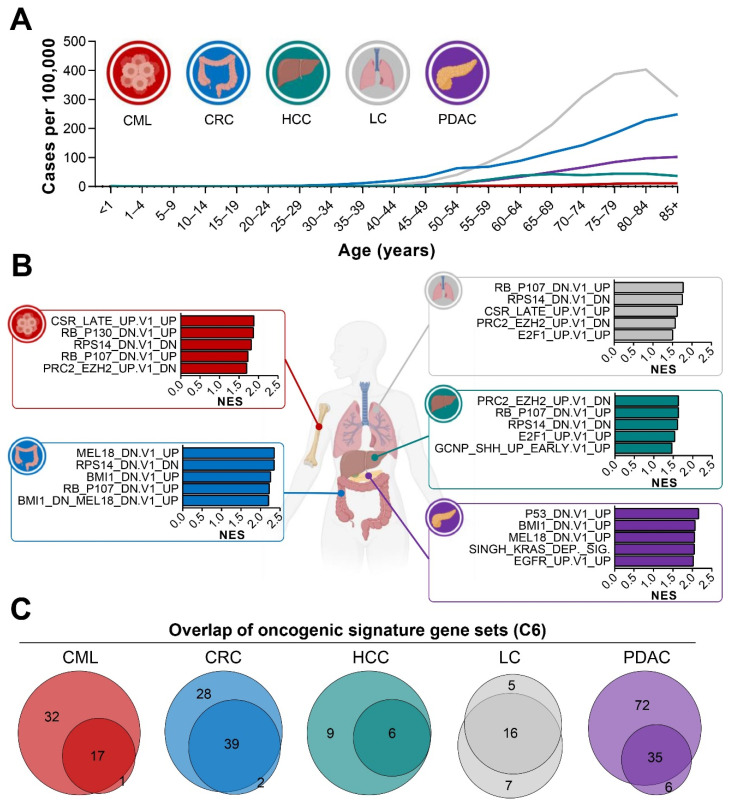
Cancer incidence increases with age and mRNA-seq analyses reveal molecular pathways underlying this group of diseases. (**A**) Cancer cases per 100,000 were obtained from publicly available sources [60] and depicted for different age groups in CML, CRC, HCC, LC, and PDAC. While global incidence was highly heterogeneous among cancer entities, all diseases were more abundant in the elderly compared to the young population. (**B**) To examine transcriptome-wide changes, GSEA for oncogenic signatures (Collection 6: oncogenic signature gene set, [C6]) was performed using publicly available CML, CRC, HCC, LC, and PDAC mRNA-seq datasets [34,36,38,41,45]. Five pathways with the highest enrichment were exemplarily shown for one dataset per cancer entity. Raw data is provided in Appendix A. (**C**) A comparison of enriched GSEA oncogenic signatures (C6) revealed high similarities between the two datasets analyzed per cancer type. Raw data is provided in Appendix A. Parts of panel A and B were created with BioRender.com (accessed on 14 September 2021). NES: normalized enrichment score, CML: chronic myelogenous leukemia, CRC: colorectal cancer, HCC: hepatocellular carcinoma, LC: lung cancer, PDAC: pancreatic ductal adenocarcinoma.

**Figure 2 cells-10-03126-f002:**
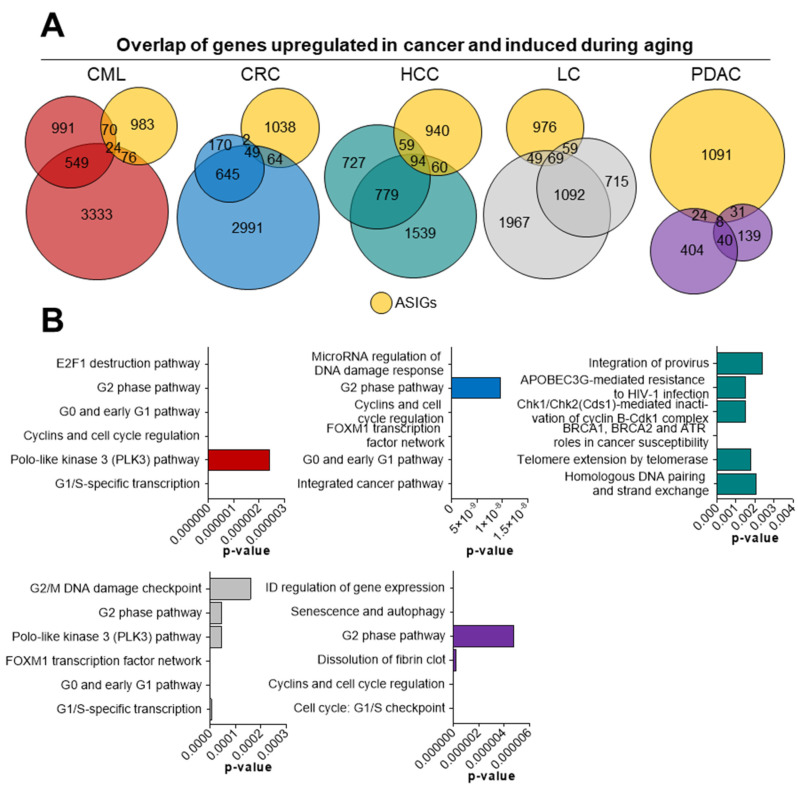
Aging/senescence-induced genes upregulated in malignant diseases. (**A**) In total, 1153 aging/senescence-induced genes (ASIGs; yellow) were defined based on previously published studies (see methods). These ASIGs were overlapped with genes upregulated in cancer bulk mRNA-seq datasets (log_2_ fold change ≥ 0.75, padj < 0.05). (**B**) ASIGs upregulated in cancer were analyzed using gene ontology (Enrichr, BioPlanet 2019 [61]).

**Figure 3 cells-10-03126-f003:**
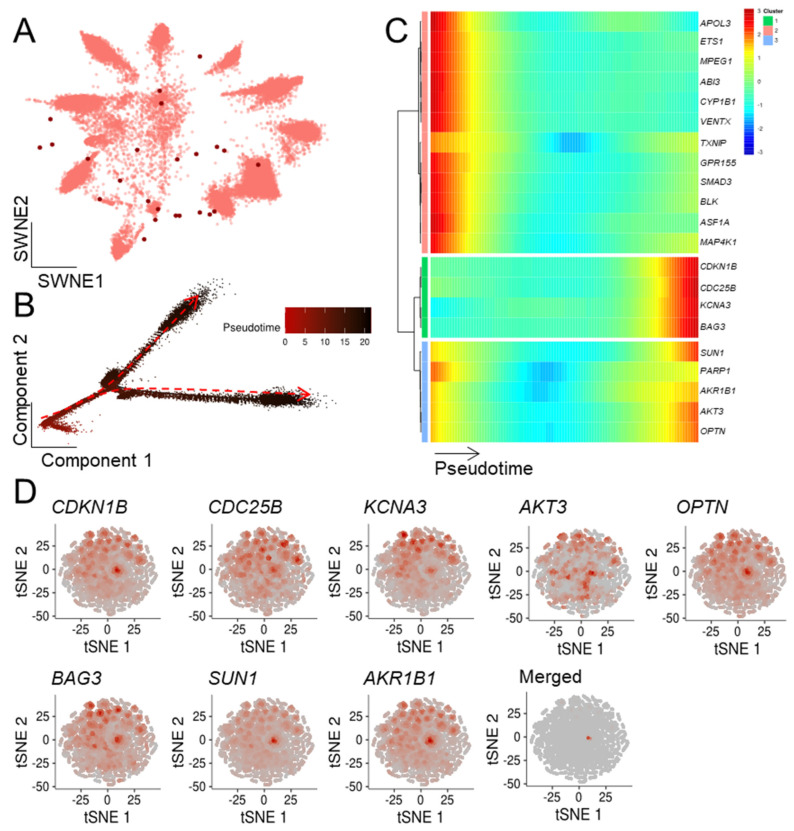
Within multiple myeloma cells, a distinct population of malignant cells displayed an upregulation of ASIGs. A publicly available malignant myeloma scRNA-seq dataset (15 patients; GSE175385 [49]) was downloaded and analyzed for transcriptional patterns of plasma cells. (**A**) The SWNE plot was weighted based on 24 previously defined, robustly expressed ASIGs (depicted as red dots). (**B**) Their developmental stage was presented in pseudotemporal ordering [55], indicating early (left, red) and late (right, black) fates. (**C**) Heatmap depicting ASIGs (green cluster) displaying significant upregulation during cellular development. (**D**) ASIGs enriched during late development were depicted in an independent tSNE representation. Merging expression patterns revealed a distinct plasma cell subpopulation with a high abundance of ASIGs.

**Figure 4 cells-10-03126-f004:**
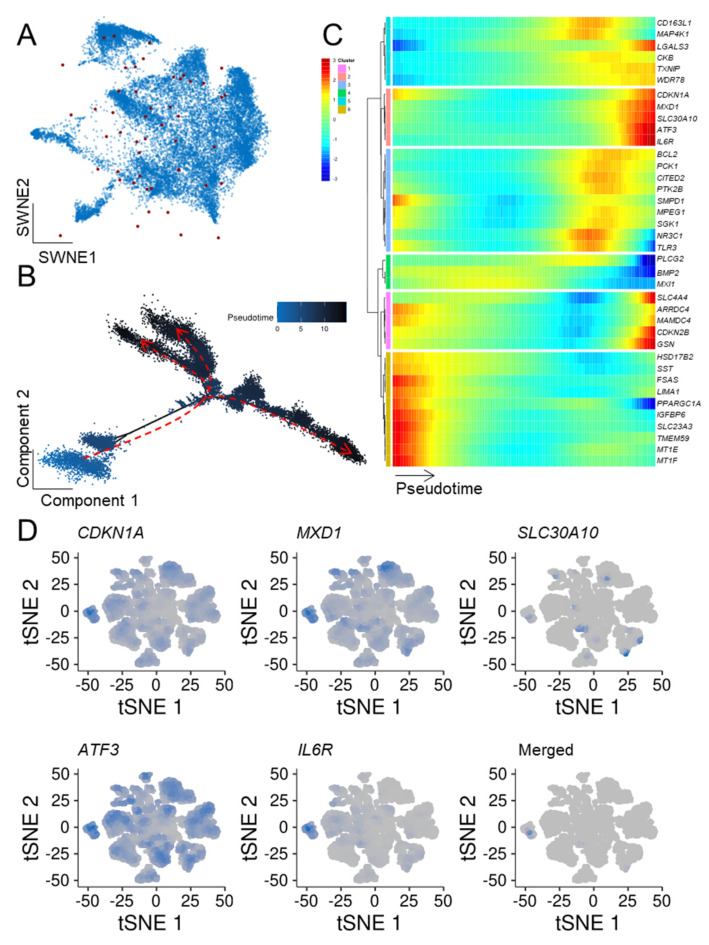
Colorectal cancer cells show a distinct population high in ASIG expression. (**A**) Epithelial cells isolated from CRC samples of 23 patients (GSE132465 [51]) were depicted in an SWNE weighted according to 49 previously defined and robustly expressed ASIGs (depicted as red dots). (**B**) Pseudotime analysis revealing early (bottom, blue) and late (top, black) fates. (**C**) Expression patterns along pseudotime indicated two clusters (orange and pink) displaying late upregulation of ASIGs. (**D**) Independent tSNE clustering of ASIGs enriched in late development phases of epithelial CRC cells.

**Figure 5 cells-10-03126-f005:**
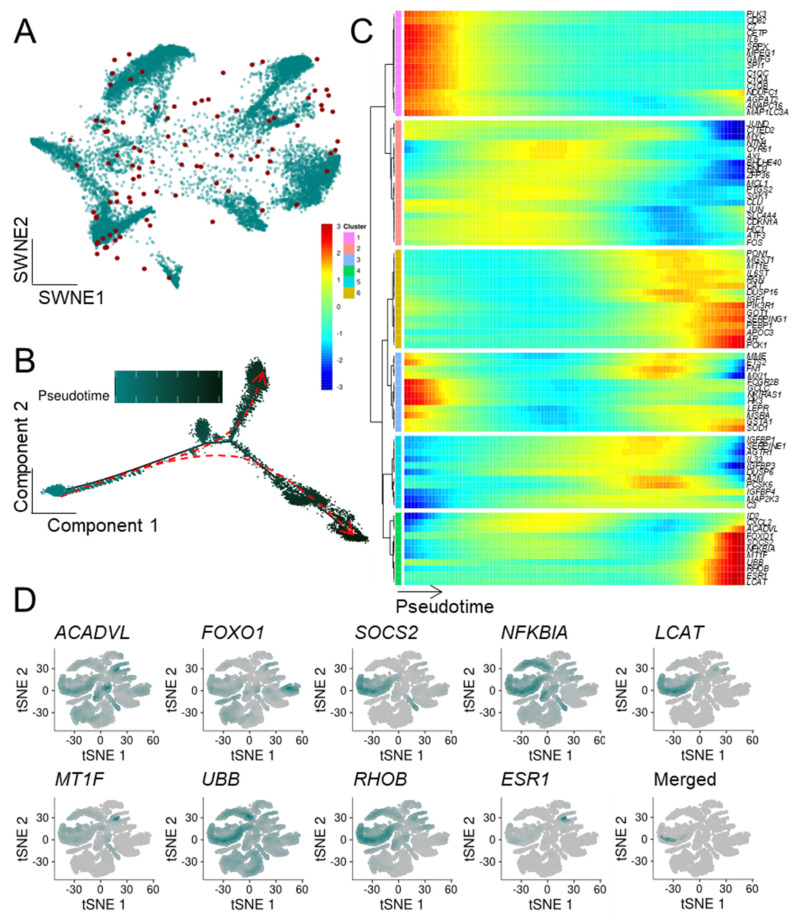
Hepatocytes display an upregulation of ASIGs in HCC. Hepatocytes isolated from a HCC scRNA-seq dataset containing samples of 46 patients were analyzed (GSE151530 [52]). (**A**) SWNE plot weighted based on 94 robustly expressed ASIGs (depicted as red dots) in HCC. (**B**) The cellular development of hepatocytes was presented in pseudotemporal ordering indicating an early (left, green) and late (right, black) fate. (**C**) Distinct gene clusters (mustard, green) showed significant upregulation of expression levels along pseudotime. (**D**) t-SNE plot displaying a cluster of cells highly expressing ASIGs genes during late development stages of hepatocytes.

**Figure 6 cells-10-03126-f006:**
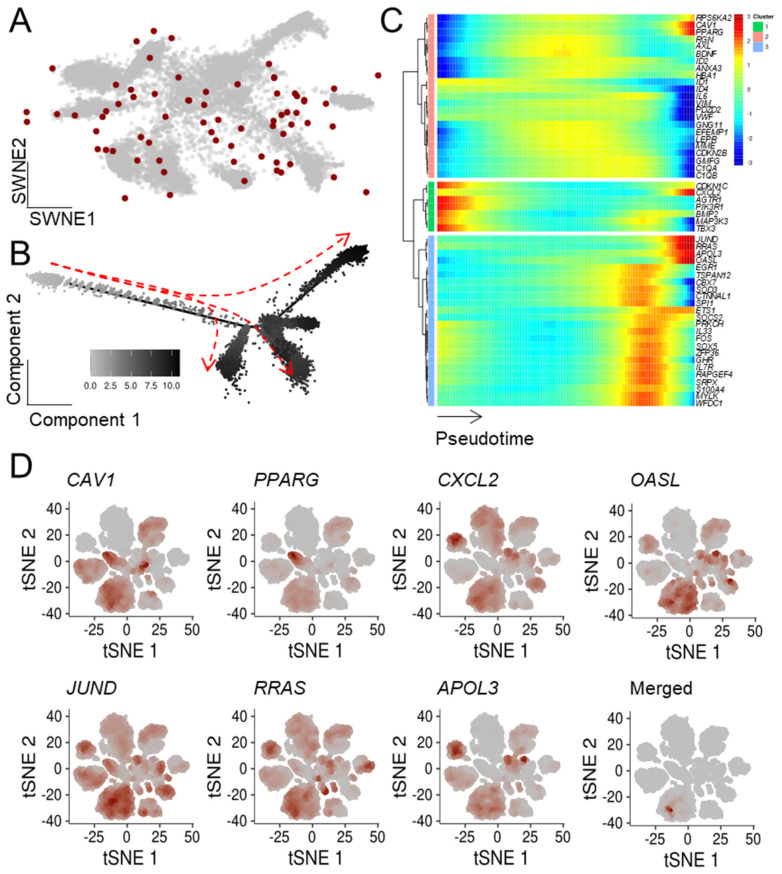
A subpopulation of lung adenocarcinoma cells displayed an upregulation of ASIGs with pseudotime. Malignant lung epithelial cells isolated from an LC scRNA-seq dataset containing samples of 11 tumors were analyzed (GSE131907 [50]). (**A**) The SWNE representation was weighted based on 69 previously defined ASIGs upregulated in LC (depicted as red dots). (**B**) Their developmental stage was depicted in pseudotemporal ordering, indicating an early (left, grey) and late (right, black) fate. (**C**) Several genes found in the blue and orange clusters displayed significant upregulation along pseudotime. (**D**) Cells expressing these genes formed a distinct cluster subpopulation as depicted in a t-SNE.

**Figure 7 cells-10-03126-f007:**
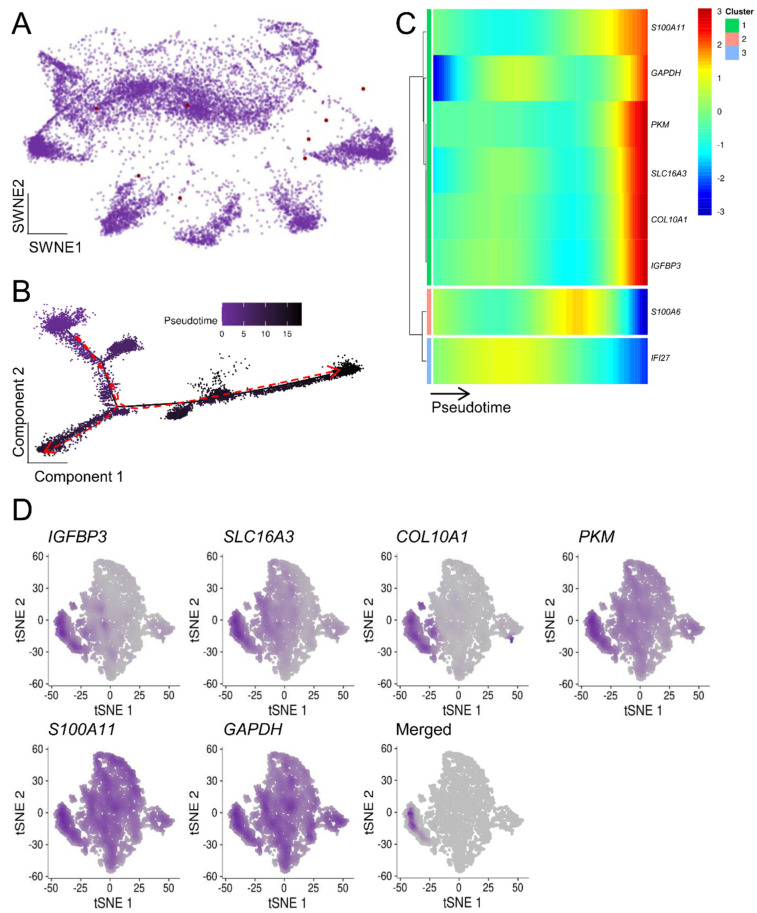
The heterogeneity of PDAC complicated defining the role of ASIGs. (**A**) In total, 24 PDAC samples (PRJCA001063 [53]) were weighted in an SWNE graph based on 8 previously defined ASIGs induced in PDAC (depicted as red dots). (**B**) The pseudotime development of malignant PDAC cells from early (left, purple) to late (right, black) stages indicated the presence of three distinct cell fates. (**C**) *S100A11, GAPDH, PKM, SLC16A3, COL10A1,* and *IGFBP3* were significantly upregulated over time. (**D**) Cells expressing these markers form a distinct subpopulation as depicted in a t-SNE.

## Data Availability

This study did not report any new data.

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
