# Peer review of "Single-Cell Transcriptomics Reveals the Expression of Aging- and Senescence-Associated Genes in Distinct Cancer Cell Populations"

_cells, 2021, doi:10.3390/cells10113126_

Round 1

Reviewer 1 Report

In this original research article, the authors investigated similarities between cancer and age-related transcriptional changes to better understand how aging could play a critical role in the development of cancer. To study this working hypothesis, the authors focused on aging-induced genes and led the authors to identify a specific set of genes. The authors also used data from various cancer types and generated using bulk or single-cell RNA-sequencing publicly available. The authors are finally describing gene and pathway overlapping between healthy aging, senescence and development of cancer. This article is using an interesting method to interpret single-cell RNA-seq which can also be applied to bulk RNA-seq. 

However, critical information is missing to allow the reproduction of the results including the exact datasets used, the strategy for excluding samples, the exact parameters used to normalize and quantify raw data and the detailed strategy to annotate the single-cell datasets. Similarly, reference and setup of the tools used for the analysis are missing. Finally, the description of the set of genes used for the interpretation needs to be performed in more detail.

1)    L77: it is unclear what are the seventeen aging signatures, are they the dataset used for building the gen set? I could only count 16 references.

2)    Could the authors comment why they decided to not rank the gene relevance? Simple strategies, such as the frequency genes appearing in different databases, or more complex strategies might have significantly improved the results. However I also understand that testing and implementing such methodology might be out of the scope of this study but need to be at least discussed.

3)    A detailed analysis of the gene overlapping between different databases and with different annotations (cell aging, senescence, cancer…) would help the reader to better understand which genes were selected, how heterogeneous the reference databases and the complexity of the gene list. Similarly, it would be interesting to also describe how many of these genes should be expressed ubiquitously and which one should be expressed on the relevant tissue of the study. Many complete databases, such as Gtex, could be used for this purpose and could be compared to the datasets used in this study.

4)    L93: how was exactly calculated the log2 FC and the p-value adjustment? Was any specific software or algorithm used? Similarly, how were the RNA data normalized? How heterogeneous were the datasets? Were low expressed transcripts ignored? The authors need to better describe the experimental design. Ideally a Jupyter notebook or any other way of sharing open source code would be beneficial for the reproducibility.

5)    L94: which version of GSEA was used? Could the authors provide more details?

6)    It is unclear how healthy bone marrow datasets were added. Are these datasets published or deposited? If yes, the reference needs to be included. If not, the dataset should be made available.

7)    L107: can the authors describe for the reader what is a “whole gene comparison”.

8)    L108, but also overall, it is unclear how the data deposited were reanalyzed for the analysis (normalization, statistics, annotation…)

9)    It is unclear how the authors were matching tumor/control in the different single-cell datasets.

10) Figure 1: we agree with the author that the overlap between signature genes is very small but similar between the studies. As mentioned in point 4, we are convinced that providing the code used to calculate the gene expression would allow the reader to redo the analysis. This would be better than providing the list of samples and the genes reported in the Venn diagrams which are currently missing.

11) Figure 2: overlap of AIG and gene upregulated is small (~1 to 150 on a list of >1000 genes). Were the results of the bulk RNA-seq influenced by the age of the individuals or the age of cancer onset? The authors should comment if this information was recorded in some dataset and show if the expected enrichment of genes involved in aging could be detected in the tissues studied in this article.

12) Why is figure 2 restricted to the analysis of genes upregulated? I could not find an explanation in the manuscript.

13) In figure 4 and figure 5: it is unclear based on which criteria the authors annotated a cell as malignant, similarly in HCC.

14) Pseudotimes are very heterogeneous between the different cancer types. Interpretation of the different profiles based on the cellular biology of each cancer type and cell type would be helpful to comment on the relevance of the presented data.

15) t-sne plots without any other annotation that the AIG is not very helpful for the reader.

16) In the discussion L279-281, the authors mentioned that the expression of the 1153 AIG was tested. This information would be interesting to display.

17) Figures illustrating pseudotime and gene expression are describing clusters not represented in any other graphs. Moreover, authors insist on the overexpression of AIG with development, but the opposite is also observed (figure 3C, cluster1). The authors need to better explain what are the different clusters and how do they match to the cancer type studied.

18) Can the authors explain why PDAC presented less AIG than other cancers? Could it be because the datasets are more heterogeneous? Or is it related to the tumor biology?

To conclude, the aim of the study is overstated. The authors are writing that the described findings can be used as therapeutic targets. But the submitted descriptive study should rather be communicated as an interesting methodological article that can be used by the community to integrate datasets. The value of this work is still high for highlighting available datasets and allowing the development of complementary data analysis or test hypotheses on the role of aging in the development of specific tumors.

Author Response

In this original research article, the authors investigated similarities between cancer and age-related transcriptional changes to better understand how aging could play a critical role in the development of cancer. To study this working hypothesis, the authors focused on aging-induced genes and led the authors to identify a specific set of genes. The authors also used data from various cancer types and generated using bulk or single-cell RNA-sequencing publicly available. The authors are finally describing gene and pathway overlapping between healthy aging, senescence and development of cancer. This article is using an interesting method to interpret single-cell RNA-seq which can also be applied to bulk RNA-seq. 

However, critical information is missing to allow the reproduction of the results including the exact datasets used, the strategy for excluding samples, the exact parameters used to normalize and quantify raw data and the detailed strategy to annotate the single-cell datasets. Similarly, reference and setup of the tools used for the analysis are missing. Finally, the description of the set of genes used for the interpretation needs to be performed in more detail.

1)    L77: it is unclear what are the seventeen aging signatures, are they the dataset used for building the gen set? I could only count 16 references.

We apologize for the lack of clarity. The seventeen aging signatures are lists of aging/senescence-associated genes which were experimentally collected and previously published. In order to clarify this for the readers, we added following text to the manuscript: “In order to define aging-associated expressional patterns, we combined seventeen previously established and experimentally verified aging signatures: CellAge and GenAge [30], Casella et al. 2019 [31], SENESCopedia by Jochems et al. [29], Hernandez-Segura et al. 2017 [32], Magalhaes aging up [12], Fridman senescence up [14] and Purcell et al. 2014 [33] as well as following MSigDB gene sets (© Broad Institute, Inc., Massachusetts Institute of Technology, and Regents of the University of California): Biocarta longevity pathway (M13158), GOBP cell aging (M14701), Reactome cellular senescence (M27188), Tang senescence Tp53 targets up (M11850), WP tca cycle in senescence (M40058), WP senescence and autophagy in cancer (M39619), GOBP regulation of cell aging (M16568), GOBP positive regulation of cell aging (M24705) and GOBP replicative senescence (M14683).”. The reviewer is right, for the 17 gene lists we used, only 16 references were cited. “CellAge” and “GenAge” are two independent gene sets which, however, can be found in the same reference.

2)    Could the authors comment why they decided to not rank the gene relevance? Simple strategies, such as the frequency genes appearing in different databases, or more complex strategies might have significantly improved the results. However I also understand that testing and implementing such methodology might be out of the scope of this study but need to be at least discussed.

We thank the reviewer for this helpful suggestion. Indeed, an initial ranking was performed by selecting significantly regulated genes (log2FC ≥0.75 and padj <0.05). While preparing this study, we discussed whether selecting the “most important” genes based on the highest log2FC would be a better approach. However, after detecting that the two RNA-seq datasets per cancer entity displayed a substantial overlap, we assumed that further selection could shift the results to one of the two datasets. To further explain this to the reader, we added following statement to the methods section: “These criteria were used consistently, and no further ranking has been used to restrict the results in order to not take advantage of single datasets.”.

3)    A detailed analysis of the gene overlapping between different databases and with different annotations (cell aging, senescence, cancer…) would help the reader to better understand which genes were selected, how heterogeneous the reference databases and the complexity of the gene list. Similarly, it would be interesting to also describe how many of these genes should be expressed ubiquitously and which one should be expressed on the relevant tissue of the study. Many complete databases, such as Gtex, could be used for this purpose and could be compared to the datasets used in this study.

We thank the reviewer for the idea to make our gene lists more transparent. In the new Supplementary Table S2 we show the enrichment of C6 oncogenic signatures in all RNA-seq datasets, in Supplementary Table S4 the parameters of all used seventeen aging/senescence gene sets, and in Supplementary Table S5 the enrichment of these gene sets in all RNA-seq datasets. In addition, to clarify the methodology behind the RNA-seq analysis performed in this manuscript, we provide an R notebook with the respective code (PDAC_R_notebook.Rmd). Furthermore, we would like to thank the reviewer for the excellent suggestion to make use of the GTEx Multi Gene Query, and explored the appearance of each tissue-specific ASIGs for all of the five organs (colon is referred to as colon – transverse and colon - sigmoid). Please find these in Revision Figure 2 (see below).

Revision Figure 2: ASIGs and their expression in a tissue-specific manner, provided by the GTEx Portal. The expression levels of ASIGs induced in both bulk RNA-seq datasets analyzed per cancer entity was displayed. The data used for the analyses described in this manuscript were obtained from the GTEx Portal on 10/14/21). Parts of this illustration were created with BioRender.com

4)    L93: how was exactly calculated the log2 FC and the p-value adjustment? Was any specific software or algorithm used? Similarly, how were the RNA data normalized? How heterogeneous were the datasets? Were low expressed transcripts ignored? The authors need to better describe the experimental design. Ideally a Jupyter notebook or any other way of sharing open source code would be beneficial for the reproducibility.

We apologize for the missing clarity. We modified the materials and methods section describing our approach of analysing the mRNA-seq results. We added: “Quality control of fastq files was carried out via FastQC and reads were mapped to the human reference genome GRCh38.p10 using HISAT2 (version 2.0.3.3) on Galaxy [46]. Read count files were generated using the featureCounts tool [version 1.4.6.p5] and normalized as analyzed for differential gene expression, using DESeq2 for the PDAC gene set without provided raw counts (Soren Müller) [version 2.11.40.6]. For the other gene sets, the raw counts were first converted into a matrix (DESeqDataSetFromMatrix), before DESeq2 (1.32.0) was used. The differential expression (DE) analysis was likewise performed with DESeq2 (lfcThreshold=0, alpha=0.1, minimum count=0.5). Significantly differentially regulated genes were selected by a Benjamini–Hochberg adjusted p-value <0.05 and log2-fold changes ≥0.75. For pairwise dataset comparisons, we focused on upregulated genes in order to track their enrichment in cancer. The upregulated genes were selected by applying a Benjamini–Hochberg adjusted p-value <0.05 and log2-fold changes above 0.75. An exemplary RNA-seq analysis vignette was provided as a R notebook (PDAC_R_notebook.Rmd). “.

5)    L94: which version of GSEA was used? Could the authors provide more details?

We apologize for not addition the version of GSEA to the original manuscript and added “GSEA, v. 4.1.0” to the methods section.

6)    It is unclear how healthy bone marrow datasets were added. Are these datasets published or deposited? If yes, the reference needs to be included. If not, the dataset should be made available.

We thank the reviewer for this question. While there were no healthy bone marrow samples in the bulk RNA-Seq dataset, PBMCs from healthy donors were analyzed in the original manuscript. They are deposited publicly at GSE100026 and GSE144119. To clarify this aspect for the readers, we added following sentences to the materials and methods section: “In the first CML dataset (GSE100026), peripheral blood mononuclear cells (PBMC) from CML patients in the chronic phase and five control samples were compared [38]. In the second CML dataset (GSM4280636), PBMCs from 16 CML patients in the chronic phase and 6 control samples were used [39].”. For the single cell RNA-Seq dataset, we used the dataset that was referenced. Indeed, the control samples are inconsistently labelled as “MB” in GSE175385. They can be directly assessed as “GSM5332301”, “GSM5332302” and “GSM5332303”. To clarify this, we added “(GSM5332301, GSM5332302 and GSM5332303)” – which are all part of the superseries GSE175385 - to the materials section.

7)    L107: can the authors describe for the reader what is a “whole gene comparison”.

We apologize for the typing error and corrected it to “whole genome comparison”.

8)    L108, but also overall, it is unclear how the data deposited were reanalyzed for the analysis (normalization, statistics, annotation…)

We thank the reviewer for the opportunity to clarify the analysis approach. To address the reviewer’s concern, we now included detailed information for all of the single cell datasets which we used throughout the study. Starting with the MM dataset, we added “The annotation was provided by the authors of the original study, and only plasma cells were provided (“orig.ident”) [49].”. For the CRC dataset, we wrote: “The annotation for the CRC dataset (“Cell_type”, “Class”) was provided by the authors in the original publication where epithelial cells were classified as normal (class “Normal”) or malignant (class “Tumor”) [51].” For the HCC dataset, we included: “The annotation for the HCC dataset was provided in the original study where it was referred to as “Type” [52].”. For the LC, we included “The annotation for the LC dataset was provided by the authors of the original study (“Cell_type”, “Cell_subtype”) [50]. For the DE analysis, within the “Cell_type”: Epithelial cells, we compared the “Cell_subtype”: Malignant cells to all other cells.” and for PDAC: “For analyzing PDAC samples, the annotation was provided earlier [53] (“cluster”). We selected ductal cell type 2 as malignant cells for subsequent steps, as suggested by the authors ([53] PRJCA001063). For comparison purposes, we used ductal cell type 1 as benign comparator (“Control”: Ductal cell type 1”, “Malignant”: ”Ductal cell type 2”).”. For further details on the normalization process, we referred to a previously published study and added “Seurat normalization and clustering followed the recommendations of the Seurat package [54].”. In order to clarify the statistical analyses, we added descriptions for Seurat and pseudotime with monocle: “For all datasets, pseudotime trajectory was calculated following the recommendations by Trapnell [55]. In detail, a lower detection limit of 0.5 was selected, and dimensional reduction with “DDRTree”, a maximum of four components, no increase of expression values before dimensional reduction (pseudo-expression=0) and scaling of each gene before trajectory reconstruction (scaling= TRUE). The differential gene test was performed according to the formula “~sm.ns(Pseudotime)”, with relative expression values (relative_expr=TRUE). Subsequently, just the significant genes were subset (qval < 0.1) and plotted on the heatmap.”.

9)    It is unclear how the authors were matching tumor/control in the different single-cell datasets.

We apologize for the unclarity and will address this issue separately for all datasets below. 
For the MM dataset, we added “Subsequently, healthy plasma cells were used as controls for comparison purposes, and “MM” samples as malignant samples.”.
For the CRC dataset, the “class” was provided as being “Tumor” or “Normal” in the original study, so we added following sentence: “The annotation for the CRC dataset (“Cell_type”, “Class”) was provided by the authors in the original publication where epithelial cells were classified as normal (class “Normal”) or malignant (class “Tumor”) [51].”.
For the HCC dataset, the types “B cells”, “CAFs”, “Malignant cells”, “T cells”, “TAMs”, “TECs”, and “unclassified” were provided in the original study. “Malignant cells” were defined and compared to all other cell types (“Other”) combined. In the manuscript, we already provided this information, but added (“Other”) and (“Malignant cells”) for clarification.
For LC, we added “or the DE analysis, within the “Cell_type”: Epithelial cells, we compared the “Cell_subtype”: Malignant cells to all other cells.”.
For PDAC, we added We selected ductal cell type 2 as malignant cells for subsequent steps, as suggested by the authors ([53] PRJCA001063). For comparison purposes, we used ductal cell type 1 as benign comparator (“Control”: Ductal cell type 1”, “Malignant”: ”Ductal cell type 2”).” for clarification.
For the technical description of the comparisons, we added following explanation: “Comparisons between control and malignant cells were calculated with the “Comparisons between control and malignant cells were calculated with the “FindMarkers” function (Seurat package), while using the “MAST” package [56] (FindMarkers(ident.1=”Control”, ident.2=”Malignant”, test.use=”MAST”, min.cells.feature=3, min.cells.group=3, min.pct=0.1, logfc.threshold=0.25).”.

10)    Figure 1: we agree with the author that the overlap between signature genes is very small but similar between the studies. As mentioned in point 4, we are convinced that providing the code used to calculate the gene expression would allow the reader to redo the analysis. This would be better than providing the list of samples and the genes reported in the Venn diagrams which are currently missing.

We agree with the reviewer that the addition of the RNA-Seq vignette (methods: PDAC_R_notebook.Rmd) makes the analysis and results more transparent. For this purpose, we provided an explanatory html script (PDAC_R_notebook.nb.html).

11)    Figure 2: overlap of AIG and gene upregulated is small (~1 to 150 on a list of >1000 genes). Were the results of the bulk RNA-seq influenced by the age of the individuals or the age of cancer onset? The authors should comment if this information was recorded in some dataset and show if the expected enrichment of genes involved in aging could be detected in the tissues studied in this article.

We thank the reviewer for this important and thoughtful question. To address this concern, we analyzed all RNA-seq patient samples (A) and scRNA-seq datasets (B) with regards to patient age and added a new supplementary figure to the manuscript (Supplementary Figure S4; see below). Unfortunately, two RNA-seq datasets (HCC GSE105130 and PDAC E-MTAB-3494) and one scRNA-seq dataset (PRJCA001063) did not provide the patients’ age. Following paragraphs were added to the results section describing the bulk RNA-seq analysis: “In addition, we compared control and tumor groups with regards to patients’ age (Supplemental Figure S4A). While two datasets did not provide this information (HCC GSE105130 and PDAC E-MTAB-3494), a minor age difference (4.4 years) occurred in one lung dataset (GSE81089). In the CML dataset (GSE100026), the 24.8 years difference did not lead to a substantially higher number of upregulated genes compared to the other CML dataset (94 genes in GSE100026 vs. 100 genes in GSE144119).”.  For the scRNA-seq analysis, we included following text: “We first compared control and malignant samples (Supplementary Figure S4B, not provided for CML GSE175385), and except for the PDAC dataset (PRJCA001063), we found no significant age differences which is why decided to compare malignant and control cells exclusively from the same sample in the PDAC dataset (ductal cell type 1 cells = control and ductal cell type 2 = malignant).”.
While we do not expect that an age difference of 4.4 years (LC) affects our findings, we added the age difference of 24.8 years in the first CML dataset to the limitations section (see below) which was now added to our manuscript. The age difference in the PDAC dataset is with 12.5 years still suboptimal but it does not appear to influence our results, since we compared ductal cell type 1 cells (as control) with the malignant ductal cell type 2 cells from the same samples for Supplementary Figure S2. 
“Limitations
While the age difference of the first LC dataset (GSE81089; 4.4 years) is unlikely to have influenced the DE results, the age difference in the first CML dataset (GSE100026) is with 24.8 years substantial and might have affected the overlap with ASIGs. However, the overlap with the ASIGs between both CML datasets was comparable (94 genes in GSE100026 vs. 100 genes in GSE144119). In the PDAC scRNA-seq dataset (PRJCA001063), the control patients were substantially younger compared to the tumor patients, which is why we decided to compare the ductal cell type 1 (control) vs. ductal cell type 2 (malignant) cells from the same samples for the single cell analysis (Supplementary Figure S2). In addition, it should be noted that next generation sequencing analyses simplify the complex biological behavior of cancers and individual cancer cells while important interactors and their influence on the cancer cell are not taken into account.”.

Supplementary Figure S4: Patients’ age in bulk RNA-seq and scRNA-seq studies. (A) Out of the publicly available bulk RNA-seq datasets analyzed in this manuscript, two studies did not provide patients’ age (paired samples in HCC GSE105130, unpaired samples in PDAC E-MTAB-3494). For the first LC dataset (GSE81089; the difference was 4.4 years between the cohorts: 63.53 years in control vs. 67.93 years in tumor, Mann-Whitney test p=0.0079). In the first CML dataset, there was a 24.8-year difference (GSE100026; control =25 years vs. tumor = 49.8 years, Mann-Whitney test=0.0079). In the CML (GSE144119) and CRC (GSE50760) datasets, the range instead of the single patient characteristics was provided (leading to a depiction as bar graph with median). (B) In the single cell datasets, the samples in control and tumor were of comparable age, except for the PDAC dataset (PDAC PRJCA001063, control = 47.0 years vs. tumor=59.54 years, unpaired t test, p=0.0015). For the CML dataset (GSE175385), the age was not provided. D'Agostino & Pearson test for normality. If passed, an unpaired t-test was performed. Otherwise, a Mann-Whitney test was performed. *p<0.05, **p<0.01, ***p<0.001.

12)    Why is figure 2 restricted to the analysis of genes upregulated? I could not find an explanation in the manuscript.

We thank the reviewer for this question and apologize for not clarifying this important aspect in our manuscript. Our decision to investigate upregulated genes was based on chromatin immunoprecipitation (ChIP) sequencing approaches detecting the occupancy of acetylated lysine 27 of histone 3 (H3K27ac), a marker for active promoters and enhancers indicating gene expression levels (Roadmap Epigenomics Consortium. Nature. 2015). It was demonstrated in several cancer entities that H3K27ac was increased in tumor samples compared to controls (e.g. Pomerantz et al. Nat. Genet. 2020, Li et al. Clin Epigenetics. 2019). These finding suggested the importance of the induction of distinct genes during cancer development and/or progression.  Therefore, we decided to focus on the concordant enrichment of genes in cancer and aging. The introduction contains following paragraph to explain our approach: “Similar to aging, extensive research has been performed in order to identify transcriptome-wide changes underlying malignant transformation in different tissues. As revealed using chromatin immunoprecipitation (ChIP) sequencing, the levels of acetylated lysine 27 of histone 3 (H3K27ac), a marker for active promoters and enhancers indicating gene expression levels [22], was increased in tumor samples compared to controls [23,24].”. In addition, to focus the reader, and not lose stringency, we added following sentences to the methods: “For pairwise dataset comparisons, we focused on upregulated genes in order to track their enrichment in cancer. The upregulated genes were selected by applying a Benjamini–Hochberg adjusted p-value <0.05 and log2-fold changes above 0.75.”.

13)    In figure 4 and figure 5: it is unclear based on which criteria the authors annotated a cell as malignant, similarly in HCC.

We are grateful for the opportunity to clarify this. To address this concern, we provided the annotation for “normal” and “tumor for each study separately. We now explicitly state this in the methods section for the scRNA-seq analysis. For Figure 4 (CRC), we added “The annotation for the CRC dataset (“Cell_type”, “Class”) was provided by the authors in the original publication where epithelial cells were classified as normal (class “Normal”) or malignant (class “Tumor”) [51].”. For Figure 5 (HCC), we added: “The annotation for the HCC dataset was provided in the original study where it was referred to as “Type” [52]. The HCC dataset contained malignant hepatocytes as well as tumor-associated macrophages, T cells, B cells, cancer-associated fibroblasts, tumor-associated endothelial cells and unclassified cells. All of them (“Other”) were compared to malignant hepatocytes for the overall comparison (“Malignant cells”) ([52] GSE151530).”.

14)    Pseudotimes are very heterogeneous between the different cancer types. Interpretation of the different profiles based on the cellular biology of each cancer type and cell type would be helpful to comment on the relevance of the presented data.

We thank the reviewer for this suggestion. We agree that pseudotime analyses are highly heterogenous between the different cancer entities tested in our study. To address the reviewer’s concern, we added following paragraph to the discussion section: “As described earlier, pseudotime analysis displays a valuable tool to explore tumor heterogeneity and to detect different intratumoral cellular states based on gene expression patterns [75]. In our pseudotime graphs we detected between two and four cellular subpopulations per cancer entity which suggests a certain degree of tumor heterogeneity. Indeed, other studies have performed in-depth evaluation of intratumoral heterogeneity using pseudotime in the past [75,76]. Future studies analyzing highly purified cell populations combined with functional analyses could help to further describe intratumoral cell populations in the datasets used throughout this manuscript.”. 

15)    t-sne plots without any other annotation that the AIG is not very helpful for the reader.

We agree with the reviewer. For this reason, we decided to not show an overall t-SNE, but a SWNE instead. With the matrix factorization used in this package, key features can be visualized more easily. While single features can readily be color-coded (like in the t-SNE depiction that we stepwise merge), the SWNE can show the key genes and indicate their expressional pattern (Revision Figure 1; see below). We exemplarily depicted this in the Revision Figure 1, where B exemplarily shows the increased expression of IGFBP3 “around” the key marker “RBP2” (right) and C exemplarily the expression of IFI27 around this specific area (left).

Revision Figure 1: Pairwise comparison of SWNE plot and underlying expressional patterns. (A) The overall SWNE is depicted, while an increased expression of (B) IGFBP3 and (C) IFI27 is shown near the respective red dots, marked in A.

16)    In the discussion L279-281, the authors mentioned that the expression of the 1153 AIG was tested. This information would be interesting to display.

We thank the reviewer for the opportunity to clarify this. All ASIGs were analyzed using bulk RNA-seq in Figure 2A and on single cell level in two respective clusters (control vs. malignant) in Supplementary Figure S2.

17)    Figures illustrating pseudotime and gene expression are describing clusters not represented in any other graphs. Moreover, authors insist on the overexpression of AIG with development, but the opposite is also observed (figure 3C, cluster1). The authors need to better explain what are the different clusters and how do they match to the cancer type studied.

We thank the reviewer for this question. It is indeed interesting that the ASIGs for each cancer entity do not behave uniformly in pseudotime. The technical potential of pseudotime enables us to form distinct clusters along the cells’ individual development. In Figure 3C, for example, some expression patterns are highest in the beginning of cell development (cluster 1), while other genes are expressed predominantly during terminal differentiation (cluster 2, partly cluster 3). Interestingly, the genes which are acquired through pseudotime are likewise common in certain clusters which form distinct populations when depicted in a tSNE (Figure 3D). A biological reason for that is described in the discussion. We additionally added “The enrichment of certain ASIGs along pseudotime and the accumulation of these genes in distinct cellular subpopulations as shown in a tSNE reflect the presence of an aging-associated subpopulation in MM cancer cells. These cells might display a potential target for specific senolytic approaches, as discussed by Carpenter et al [83].”.

18)    Can the authors explain why PDAC presented less AIG than other cancers? Could it be because the datasets are more heterogeneous? Or is it related to the tumor biology?

We were surprised by this outcome as well and hypothesized that there might be a high degree of heterogeneity in the PDAC samples. However, after obtaining the reviewer comments, we re-analyzed all samples. Surprisingly, we discovered that one sample was mislabeled (E-MTAB-3494 was stated as “raw counts”, while it was “normalized counts”). Subsequently, we reanalyzed all PDAC samples, leading to a higher enrichment of AIGs (now referred to as ASIGs based on the suggestions of reviewer #1). Accordingly, all figures and files provided with this manuscript were changed. We apologize for not noticing this mistake earlier.

To conclude, the aim of the study is overstated. The authors are writing that the described findings can be used as therapeutic targets. But the submitted descriptive study should rather be communicated as an interesting methodological article that can be used by the community to integrate datasets. The value of this work is still high for highlighting available datasets and allowing the development of complementary data analysis or test hypotheses on the role of aging in the development of specific tumors.

We are grateful for the reviewer’s constructive feedback and suggestion. We agree with the reviewer that rather than indicating therapeutic targets, this article can serve as a blueprint on how to analyze existing datasets in order to complement the role of aging in cancer. To address the reviewer’s concern, we deleted the final sentence of the discussion and added following sentence: “The data presented here can serve as a basis for future studies on the role of aging in the development of specific tumors and be used to integrate further datasets for complementary analyses.”. In addition, we changed the title to “Single-cell transcriptomics reveals the expression of aging- and senescence-associated genes in distinct cancer cell populations”. We thank the reviewer for all of their really helpful suggestions which helped us to improve the manuscript substantially.

Reviewer 2 Report

In the manuscript “Transcriptome-wide profiling of cancer and aging”, Authors address a current topic, which is the role of aging–regulated genes on cancer development. In more detail, the manuscript presents a meta-analysis based on publicly available data (bulk RNAseq and single cell RNAseq) for different cancers (chronic myelogenous leukemia, colorectal cancer, hepatocellular carcinoma, lung cancer and pancreatic ductal adenocarcinoma). The idea is to verify and analyze the relationship between gene expression regulation in aging and cancer.

Overall, the manuscript is interesting but, in the present form, is not fully comprehensible, mainly due to the lack of detailed Methods section. Also Discussion is weak, and the novelty of the manuscript is modest.  

An itemized, point by point comment follows:

  1. I strongly suggest to modify the title of the paper, since the current one is too generic.
  2. Authors refer to “Aging-Induced Genes” throughout the manuscript. They report to use a list of aging-associated expressional patterns obtained by combining “seventeen widely used aging signatures” (references 12, 14, 26-30, and reported gene sets- see manuscript lines 82-86). The term “aging” is misleading, since it is not synonym of “senescence”. Most of the cited references and gene sets are focused on genes regulated in cell senescence rather than in organismal aging.
  3. Authors use a list of 1,153 aging-induced genes obtained combining data from seventeen literature genes (those reported in the previous point). My question is: since Authors refer to aging-induced genes, did they select only upregulated genes? In this case, it should be clearly stated and a comment about the rationale of considering only upregulated genes should be provided. Furthermore, it is necessary to report the cut off by which genes were selected, in terms of fold change increase.
  4. Materials and Methods, “Aging signatures” sub-heading: please summarize the aging signatures in a table.
  5. Materials and Methods, “Aging signatures” sub-heading: Reference 12 is Magalhaes (not Demagalhaes)
  6. Materials and Methods, “Aging signatures” sub-heading: If I understand well, Authors take into account all significantly modulated genes in the available bulk mRNA sequencing data (log2 fold change >0.75 or <0.75). Figure 2A reports Venn diagrams showing AIG overlapping with “genes upregulated in cancer”. Why do Authors select a list of significantly downregulated genes in cancer data sets if this group of genes in not considered in the following analysis?
  7. Materials and Methods, “Analysis of the bulk mRNA sequencing data” sub-heading: similarly to point 4, I suggest to summarize the datasets for each cancer type in a table.
  8. Materials and Methods, “Single cell-RNAseq (scRNA-seq analysis) sub-heading: tumor acronyms have already been used in the manuscript: use only the abbreviations (see lines 98-99).
  9. Results section, lines 136-138 and Figure 1A: please, list the tumor types in ascending or descending order (fold increase of tumor incidence with age) in the text and in the figure (tumor icons).
  10. Figure 1B: Please explain more clearly the signatures emerging from GSEA and explain the role of the enriched pathways in the different tumors. In legend to Fig. 1B, explain what C6 and NES stand for.
  11. Lines 163-164: please order (ascending or descending order) the list of AIGs (i.e. only 10 AIGs were upregulated in PDAC, 115 in…, 170 in..…177 in…and 213 in…).
  12. Results section, line 169: it is not clear to me which are the healthy control samples: every cancer tissue was compared to the healthy tissue coming from the same patient? Please explain.
  13. Lines 192-193: please explain the characteristics of SWNE and of pseudotime analysis. The manuscript is directed to an audience who might likely be not familiar with these analyses.
  14. scRNA seq analysis: I have a comment on the use of single cell analysis. In general, although this approach is interesting and provides data about regulation of gene expression of the single cancer cell which are interesting, it necessarily excludes from the analysis important interactors of tumor cells, providing a partial perspective of this complex scenario.

Author Response

In the manuscript “Transcriptome-wide profiling of cancer and aging”, Authors address a current topic, which is the role of aging–regulated genes on cancer development. In more detail, the manuscript presents a meta-analysis based on publicly available data (bulk RNAseq and single cell RNAseq) for different cancers (chronic myelogenous leukemia, colorectal cancer, hepatocellular carcinoma, lung cancer and pancreatic ductal adenocarcinoma). The idea is to verify and analyze the relationship between gene expression regulation in aging and cancer.
Overall, the manuscript is interesting but, in the present form, is not fully comprehensible, mainly due to the lack of detailed Methods section. Also Discussion is weak, and the novelty of the manuscript is modest.  
An itemized, point by point comment follows:

  1.    I strongly suggest to modify the title of the paper, since the current one is too generic.

We thank the reviewer for this suggestion. After substantially refining the focus of this manuscript based on the reviewers’ recommendations, we decided to change the title of this study accordingly. The new title is “Single-cell transcriptomics reveals the expression of aging- and senescence-associated genes in distinct cancer cell populations”.

  1.    Authors refer to “Aging-Induced Genes” throughout the manuscript. They report to use a list of aging-associated expressional patterns obtained by combining “seventeen widely used aging signatures” (references 12, 14, 26-30, and reported gene sets- see manuscript lines 82-86). The term “aging” is misleading, since it is not synonym of “senescence”. Most of the cited references and gene sets are focused on genes regulated in cell senescence rather than in organismal aging.

We agree with the reviewer and thank them for pointing out this aspect. Indeed, the majority of genes upregulated during the process of aging are associated with senescence. As mentioned by the reviewer, aging and senescence, however, should not be generalized and used as synonyms. In order to address this concern and for the sake of clarity, we replaced the acronym AIG (aging-induced gene) which was used throughout the manuscript by ASIG (aging/senescence-induced gene). In addition, we briefly explained the acronym by adding following sentence to the materials and methods section: “Due to their upregulation during cellular aging and senescence, these genes were referred to as aging/senescence-induced genes (ASIGs) throughout this manuscript [8,9].”.

  1.    Authors use a list of 1,153 aging-induced genes obtained combining data from seventeen literature genes (those reported in the previous point). My question is: since Authors refer to aging-induced genes, did they select only upregulated genes? In this case, it should be clearly stated and a comment about the rationale of considering only upregulated genes should be provided. Furthermore, it is necessary to report the cut off by which genes were selected, in terms of fold change increase.

We thank the reviewer for this question and apologize for not clarifying this important aspect in our manuscript. Our decision to investigate upregulated genes was based on chromatin immunoprecipitation (ChIP) sequencing approaches detecting the occupancy of acetylated lysine 27 of histone 3 (H3K27ac), a marker for active promoters and enhancers indicating gene expression levels (Roadmap Epigenomics Consortium. Nature. 2015). It was demonstrated in several cancer entities that H3K27ac was increased in tumor samples compared to controls (e.g. Pomerantz et al. Nat. Genet. 2020, Li et al. Clin Epigenetics. 2019). These finding suggested the importance of the induction of distinct genes during cancer development and/or progression.  Therefore, we decided to focus on the concordant enrichment of genes in cancer and aging. The introduction contains following paragraph to explain our approach: “Similar to aging, extensive research has been performed in order to identify transcriptome-wide changes underlying malignant transformation in different tissues. As revealed using chromatin immunoprecipitation (ChIP) sequencing, the levels of acetylated lysine 27 of histone 3 (H3K27ac), a marker for active promoters and enhancers indicating gene expression levels [22], was increased in tumor samples compared to controls [23,24].”. In addition, to focus the reader, and not lose stringency, we added following sentences to the methods: “For pairwise dataset comparisons, we focused on upregulated genes in order to track their enrichment in cancer. The upregulated genes were selected by applying a Benjamini–Hochberg adjusted p-value <0.05 and log2-fold changes above 0.75.”.

  1.    Materials and Methods, “Aging signatures” sub-heading: please summarize the aging signatures in a table.

We thank the reviewer for this suggestion. All aging signatures have been added as Supplementary Table S4 which referred to in the text as follows: “All gene lists are supplied in Supplementary Table S4.”.

  1.    Materials and Methods, “Aging signatures” sub-heading: Reference 12 is Magalhaes (not Demagalhaes)

We apologize for this mistake and changed the author’s name accordingly.

  1.    Materials and Methods, “Aging signatures” sub-heading: If I understand well, Authors take into account all significantly modulated genes in the available bulk mRNA sequencing data (log2 fold change >0.75 or <0.75). Figure 2A reports Venn diagrams showing AIG overlapping with “genes upregulated in cancer”. Why do Authors select a list of significantly downregulated genes in cancer data sets if this group of genes in not considered in the following analysis?

We apologize for the lack of clarity. Indeed, we exclusively focused on the upregulated genes. The description of our approach was changed as follows: “Significantly differentially regulated genes were selected by a Benjamini–Hochberg adjusted p-value <0.05 and log2-fold changes ≥0.75.”

  1.    Materials and Methods, “Analysis of the bulk mRNA sequencing data” sub-heading: similarly to point 4, I suggest to summarize the datasets for each cancer type in a table.

We are grateful for the reviewer’s suggestion. We added Supplementary Table S2 where for each cancer type and dataset, the respective NOM p-val and NES can be found. In addition, we provide the enrichment with NOM p-val and NES for each of the seventeen aging genesets in Supplementary Table S5. In the results section, we additionally provided “The enrichment of the respective oncogenic signatures is listed in Supplementary Table S2.” and “[…,], Supplementary Table S5 […]”.

  1.    Materials and Methods, “Single cell-RNAseq (scRNA-seq analysis) sub-heading: tumor acronyms have already been used in the manuscript: use only the abbreviations (see lines 98-99).

We thank the reviewer and deleted these parts.

  1.    Results section, lines 136-138 and Figure 1A: please, list the tumor types in ascending or descending order (fold increase of tumor incidence with age) in the text and in the figure (tumor icons).

We changed the order into CML, HCC, CRC, PDAC and LC, in ascending order of fold increase of tumor incidence with age in the text. We hope that the reviewer concedes that the order of cancer entities remains unchanged in the Figure 1A since this specific order was maintained throughout the entire manuscript. In all figures, we adhered to the alphabetic order of cancer types (CML, CRC, HCC, LC, PDAC) and would like to avoid having different arrangements in the individual figure panels.

  1.    Figure 1B: Please explain more clearly the signatures emerging from GSEA and explain the role of the enriched pathways in the different tumors. In legend to Fig. 1B, explain what C6 and NES stand for.

As requested by the reviewer, we added detailed explanations of our GSEA approach and respective signatures. We added following explanations to the results section: “[…] gene set enrichment analysis of oncogenic signature genes (C6) revealed an enrichment of key driver pathways in cancer samples (Figure 1B; Supplemental Figure S1A).” and “The enrichment of the respective oncogenic signatures is listed in Supplementary Table S2”. In addition, we added “NES: normalized enrichment score, CML: chronic myelogenous leukemia, CRC: colorectal cancer, HCC: hepatocellular carcinoma, LC: lung cancer, PDAC: pancreatic ductal adenocarcinoma.“ to the figure legend, as “Collection 6= oncogenic signature gene set [C6]”. To address the role of the pathways enriched in the respective datasets, we added following paragraph to the discussion section: “Notably, all datasets displayed an enrichment of polycomb group protein-associated signatures (i.e. EZH2, BMI1 and MEL18). This group of proteins consists of epigenetic repressors modulating the transcriptional landscape, thereby controlling cell differentiation and tumorigenesis in various cancer types [68,69]. In addition, Ribosomal Protein S14 (RPS14)-related pathways were enriched in the majority of mRNA-seq studies (CML, CRC, LC and HCC). This factor was demonstrated to regulate proliferation in cancer cells, for instance by regulating the activity of the tumor-promoting factor c-MYC [70,71]. Another pathway which was enriched in most datasets (CML, CRC, LC, HCC) was associated with the well-described retinoblastoma (RB) tumor suppressor. This protein was generally described as a cell cycle regulator, however, numerous studies revealed its function in cell differentiation, survival as well as epigenetic regulation in cancer cells [72–74].”. In addition, following brief description of the GSEA analysis was added to the results section: “Interestingly, all datasets displayed the enrichment of polycomb group protein-associated signatures (i.e. EZH2, BMI1 and MEL18) and the majority (CML, CRC, LC and HCC) showed an enrichment in Ribosomal Protein S14 (RPS14)- and Retinoblastoma (RB)-related pathways.”.

  1.    Lines 163-164: please order (ascending or descending order) the list of AIGs (i.e. only 10 AIGs were upregulated in PDAC, 115 in…, 170 in..…177 in…and 213 in…).

We thank the reviewer for the idea and rearranged into descending order. The new sentence is “While in total 213 genes were upregulated in HCC, 177 genes were upregulated in LC, 170 ASIGs were induced in CML, 115 in CRC and only 63 ASIGs were induced in PDAC (Figure 2A).”.

  1.    Results section, line 169: it is not clear to me which are the healthy control samples: every cancer tissue was compared to the healthy tissue coming from the same patient? Please explain.

To address the reviewer’s question, we clarified this in the materials and methods section as follows: “In the first CML dataset (GSE100026), peripheral blood mononuclear cells (PBMC) from CML patients in the chronic phase and five control samples were compared [38]. In the second CML dataset (GSM4280636), PBMCs from 16 CML patients in the chronic phase and 6 control samples were used [39].”. In addition, we provide the sample characteristics in more detail in Supplementary Figure S4.

  1.    Lines 192-193: please explain the characteristics of SWNE and of pseudotime analysis. The manuscript is directed to an audience who might likely be not familiar with these analyses.

We agree and are grateful to the reviewer for raising this point. A SWNE plot uses matrix factorization in which key features can be easier visualized. While single features can easily be color-coded (like in the t-SNE depiction that we stepwise merge), the SWNE can show the key genes and indicate their expressional pattern (Revision Figure 1). We exemplarily depicted this in Revision Figure 1 (see below), where B shows the increased expression of IGFBP3 “around” the key gene IGFBP3 (on the right side). In panel (C) we show the expression of IFI27 around the “key gene” IFI27, highest on the left side. In the respective results section, we added following explanation: “A SWNE plot uses matrix factorization to visualize key features and their influence on single cells.”. And, subsequently, for pseudotime, we included following sentence: “Pseudotime infers the ordering of cells along a progression lineage which is taken from their expressional profile related to capture time.”.

Revision Figure 1: Pairwise comparison of SWNE plot and underlying expressional patterns. (A) The overall SWNE is depicted, while an increased expression of (B) IGFBP3 and (C) IFI27 is shown near the respective red dots, marked in A.

  1.    scRNA seq analysis: I have a comment on the use of single cell analysis. In general, although this approach is interesting and provides data about regulation of gene expression of the single cancer cell which are interesting, it necessarily excludes from the analysis important interactors of tumor cells, providing a partial perspective of this complex scenario.

We definitely agree with the reviewer that single cell analysis oversimplifies the complex biology underlying cancers and their biological behavior. To address the reviewer’s concern, we added a limitation section to our manuscript containing, amongst others, following statement: “In addition, it should be noted that next generation sequencing analyses simplify the complex biological behavior of cancers and individual cancer cells while important interactors and their influence on the cancer cell are not taken into account.”. Despite the clear limitations of this approach, we are convinced that the appearance of aging-related features within distinct cancer entities displays a novel and relevant finding which will be of interest for the readership of “Cells”.

Reviewer 3 Report

This is an interesting study at the interface between cancer and aging. The multifaceted interplay between those two processes is intriguing and more intensified research is definitely justified.

However, I do have some major concerns about this study:

  1. The authors are investigating the (over)expression of aging-induced genes (AIGs) in tumor RNAseq datasets from public databases, but apparently they did not account for patient age (unless I completely missed it). Were the tumor and healthy control tissues matched somehow in terms of age?
  2. Also, the biological relevance of these findings is not very clear to me. What role could the AIG-expressing subpopulations play in the cancer process? The authors suggest a causal relationship between aging and cancer (e.g. lines 40-42), and even stipulate that the AIG-enriched cell populations could represent therapeutic targets, but in fact it could also be the other way around and these subpopulations might also develop as a result of the tumor initiation/transformation/progression. There could for instance be a link with cellular senescence, which is known as an inherent defense mechanism against developing malignancies. In this regard, it is not surprising that tumors show increased expression of aging/senescence genes. These issues should be discussed, as well as the seemingly contradicting effects of the aging-associated gene patterns (i.e. favorable vs. detrimental effect on cancer outcomes).
  3. The authors only talk about upregulated genes, while some pathways are strongly downregulated with aging, such as the insulin/IGF-1 pathway. What about these?
  4. Lines 279-281: on which part of the data is this statement based? I could not really find this in the results section and/or figures.
  5. Line 295-299: not clear (link with microRNAs?)
  6. Another general remark is that the illustrations are not always easy to read and interpret for readers not familiar with this type of analyses. The legends should provide a bit more explicit explanation and all abbreviations should be explained. A few examples:
    • Fig.1: NES? C6?
    • Fig. 1B and Fig. S1: the overlap between the different cancer types would be much clearer if the data were presented in a table, with all the occurring deregulated genes/pathways listed below each other and the p-values in columns per cancer type (left blank if not significant for that cancer type). Green panel in fig. S1B is very difficult to read!
    • Fig. 3-7: are the genes depicted as red dots in the A panels the same that are displayed in the C panels (yet the number is a bit different)?
  7. Some methodological details are missing, for instance:
    • Line 108-112: not clear what was used as the control for HCC
    • Nothing is mentioned about numbers of cells that were included in the scRNAseq data nor about relative contributions of the different donors.
    • Line 159-160: include references for the 17 studies
    • Fig. 3-7: how exactly were the 'previously defined AIGs' selected?

Author Response

This is an interesting study at the interface between cancer and aging. The multifaceted interplay between those two processes is intriguing and more intensified research is definitely justified.

We thank the reviewer for this thoughtful assessment of our work.

However, I do have some major concerns about this study:

  1.    The authors are investigating the (over)expression of aging-induced genes (AIGs) in tumor RNAseq datasets from public databases, but apparently they did not account for patient age (unless I completely missed it). Were the tumor and healthy control tissues matched somehow in terms of age?

We thank the reviewer for this important and thoughtful question. To address this concern, we analyzed all RNA-seq patient samples (A) and scRNA-seq datasets (B) with regards to patient age and added a new supplementary figure to the manuscript (Supplementary Figure S4; see below). Unfortunately, two RNA-seq datasets (HCC GSE105130 and PDAC E-MTAB-3494) and one scRNA-seq dataset (PRJCA001063) did not provide the patients’ age. Following paragraphs were added to the results section describing the bulk RNA-seq analysis: “In addition, we compared control and tumor groups with regards to patients’ age (Supplemental Figure S4A). While two datasets did not provide this information (HCC GSE105130 and PDAC E-MTAB-3494), a minor age difference (4.4 years) occurred in one lung dataset (GSE81089). In the CML dataset (GSE100026), the 24.8 years difference did not lead to a substantially higher number of upregulated genes compared to the other CML dataset (94 genes in GSE100026 vs. 100 genes in GSE144119).”. For the scRNA-seq analysis, we included following text: “We first compared control and malignant samples (Supplementary Figure S4B, not provided for CML GSE175385), and except for the PDAC dataset (PRJCA001063), we found no significant age differences which is why decided to compare malignant and control cells exclusively from the same sample in the PDAC dataset (ductal cell type 1 cells = control and ductal cell type 2 = malignant).”.

Supplementary Figure S4: Patients’ age in bulk RNA-seq and scRNA-seq studies. (A) Out of the publicly available bulk RNA-seq datasets analyzed in this manuscript, two studies did not provide patients’ age (paired samples in HCC GSE105130, unpaired samples in PDAC E-MTAB-3494). For the first LC dataset (GSE81089) the difference was 4.4 years between the cohorts: 63.53 years in control vs. 67.93 years in tumor, Mann-Whitney test p=0.0079. In the first CML dataset, there was a 24.8-year difference (GSE100026; control: 25 years vs. tumor: 49.8 years, Mann-Whitney test=0.0079). In the CML (GSE144119) and CRC (GSE50760) datasets, the range instead of the single patient characteristics was provided in the original studies (leading to a depiction as bar graph with median). (B) In the single cell datasets, the samples in control and tumor were of comparable age, except for the PDAC dataset (PDAC PRJCA001063, control: 47.0 years vs. tumor:59.54 years, unpaired t-test p=0.0015). For the CML dataset (GSE175385), the age was not provided. D'Agostino & Pearson test for normality. If passed, an unpaired t-test was performed. Otherwise, a Mann-Whitney test was performed. *p<0.05, **p<0.01, ***p<0.001.

Also, the biological relevance of these findings is not very clear to me. What role could the AIG-expressing subpopulations play in the cancer process? The authors suggest a causal relationship between aging and cancer (e.g. lines 40-42), and even stipulate that the AIG-enriched cell populations could represent therapeutic targets, but in fact it could also be the other way around and these subpopulations might also develop as a result of the tumor initiation/transformation/progression. There could for instance be a link with cellular senescence, which is known as an inherent defense mechanism against developing malignancies. In this regard, it is not surprising that tumors show increased expression of aging/senescence genes. These issues should be discussed, as well as the seemingly contradicting effects of the aging-associated gene patterns (i.e. favorable vs. detrimental effect on cancer outcomes).

We thank the reviewer for this question. Indeed, the idea of these ASIG expressing cells being potential targets of senolytic therapeutics might have been too speculative. Therefore, we deleted this part from the discussion. We carefully evaluated the favorable and detrimental effects on local cancer development or progression by adding following paragraph to the discussion section: “Whether these cells are drivers of tumor formation/progression or the result of tumor initiation/transformation remains to be elucidated. Cellular senescence has been proposed an anti-cancer mechanism for a long time due to the divisional arrest of apoptosis-resistant cancer cells [15,111]. Controversially, experimental findings suggested that senescent cells are entailing local inflammation in skin, bone liver and other tissues, allegedly promoting malignant transformation [15,112–114]. The appearance, fate as well as the secretory phenotype of senescent cells in cancer environments suggests that senolytic therapy might serve as complementary cancer therapy as proposed by Dimri and others [111,115,116]. In addition, future research could address whether healthy cells with high expression of ASIGs show a higher tendency to undergo malignant transformation than other cells.”.

  1.    The authors only talk about upregulated genes, while some pathways are strongly downregulated with aging, such as the insulin/IGF-1 pathway. What about these?

We thank the reviewer for this question and apologize for not clarifying this important aspect in our manuscript. Our decision to investigate upregulated genes was based on chromatin immunoprecipitation (ChIP) sequencing approaches detecting the occupancy of acetylated lysine 27 of histone 3 (H3K27ac), a marker for active promoters and enhancers indicating gene expression levels (Roadmap Epigenomics Consortium. Nature. 2015). It was demonstrated in several cancer entities that H3K27ac was increased in tumor samples compared to controls (e.g. Pomerantz et al. Nat. Genet. 2020, Li et al. Clin Epigenetics. 2019). These finding suggested the importance of the induction of distinct genes during cancer development and/or progression.  Therefore, we decided to focus on the concordant enrichment of genes in cancer and aging. The introduction contains following paragraph to explain our approach: “Similar to aging, extensive research has been performed in order to identify transcriptome-wide changes underlying malignant transformation in different tissues. As revealed using chromatin immunoprecipitation (ChIP) sequencing, the levels of acetylated lysine 27 of histone 3 (H3K27ac), a marker for active promoters and enhancers indicating gene expression levels [22], was increased in tumor samples compared to controls [23,24].”. In addition, to focus the reader, and not lose stringency, we added following sentences to the methods: “For pairwise dataset comparisons, we focused on upregulated genes in order to track their enrichment in cancer. The upregulated genes were selected by applying a Benjamini–Hochberg adjusted p-value <0.05 and log2-fold changes above 0.75.”.

  1.    Lines 279-281: on which part of the data is this statement based? I could not really find this in the results section and/or figures.

We apologize for the missing clarity. This statement is based on the results from line 184-186, which directly refer to Supplementary Figure S2 as indicated in the text now.

  1.    Line 295-299: not clear (link with microRNAs?)

To improve the clarity of our statement, we changed the text as follows: “Increased CDC25B [79], AKR1B1 [80] and AKT3 expression [81] were associated with poor multiple myeloma survival, indicating a detrimental effect of certain aging-associated gene patterns. On the other hand, AKT3 is a potential therapeutic candidate as shown in vitro [82].”.

  1.    Another general remark is that the illustrations are not always easy to read and interpret for readers not familiar with this type of analyses. The legends should provide a bit more explicit explanation and all abbreviations should be explained. A few examples:
  •    Fig.1: NES? C6?

We thank the reviewer for pointing our specifically which illustrations are complicated to read or interpret. To address their first concern, we added following explanations to  Figure 1 and Figure 2 “Collection 6= oncogenic signature gene set, [C6]” and “NES: normalized enrichment score, CML: chronic myelogenous leukemia, CRC: colorectal cancer, HCC: hepatocellular carcinoma, LC: lunc cancer, PDAC: pancreatic ductal adenocarcinoma.”.

  •    Fig. 1B and Fig. S1: the overlap between the different cancer types would be much clearer if the data were presented in a table, with all the occurring deregulated genes/pathways listed below each other and the p-values in columns per cancer type (left blank if not significant for that cancer type). Green panel in fig. S1B is very difficult to read!

We rearranged the results underlying Figure 1B and Supplementary Figure S1A, providing these data in Supplementary Table 2 now. Exemplarily, we show Revision Table 1 below and hope that the review is satisfied with the changes we performed. In addition, we added following sentence to the respective figure legends: “Raw data is provided in Supplementary Table 2”. 

Revision Table 1 Exemplarily demonstration of data analyses underlying Figure 1B and Supplementary Figure S1A (fully depicted as Supplementary Table 2).

C6 oncogenic signatures    NOM p-val                                    
Pathway    HCC_GSE105130    HCC_GSE148355    NSCLC_GSE81089    Lung_AC_GSE40419    CML_GSE100026    CML_GSE144119    PDAC_GSE119794    PDAC_E-MTAB-3494    CRC_GSE50760    CRC_GSE146009
E2F1_UP.V1_UP    0.0000    0.0212    0.0000    0.0000    0.0000    0.0050    0.1118    0.1011    0.0095    0.0036
PRC2_EED_UP.V1_DN    0.0000    0.3909    0.0132    0.0562    0.0000    0.0244    0.0000    0.0000    0.0000    0.0000
PRC2_EZH2_UP.V1_DN    0.0000    0.0057    0.0015    0.0000    0.0000    0.0000    0.0284    0.0041    0.0000    0.0000
RB_P107_DN.V1_UP    0.0000    0.0000    0.0000    0.0000    0.0000    0.0053    0.0016    0.0000    0.0000    0.0000
RPS14_DN.V1_DN    0.0000    0.0000    0.0000    0.0000    0.0000    0.0000    0.0946    0.1856    0.0000    0.0000
VEGF_A_UP.V1_DN    0.0000    0.0253    0.0044    0.0000    0.0080    0.0000    0.0031    0.0156    0.0000    0.0000
GCNP_SHH_UP_
LATE.V1_UP    0.0021    0.0068    0.0044    0.0000    0.0336    0.0000    0.0168    0.1526    0.0000    0.0000
GCNP_SHH_UP_
EARLY.V1_UP    0.0039    0.5763    0.0091    0.0000    0.0000    0.0025    0.0122    0.2012    0.0132    0.0000

                            p-values    Ë‚0.1    Ë‚0.05    Ë‚0.01

Fig. 3-7: are the genes depicted as red dots in the A panels the same that are displayed in the C panels (yet the number is a bit different)?

We apologize for the missing clarity. The genes we selected for the scRNA-seq analysis were the genes enriched in both bulk RNA-seq datasets as well as the ASIGs (for CML: 24, CRC: 49, HCC: 94, LC: 69 and PDAC: 8 genes; Figure 2A). Subsequently, these genes were displayed in a SWNE presentation (Figures 3-7, panel A). Out of those factors, we selected only the genes which were significantly regulated along pseudotime (panel C, qval <0.1). The number of genes displayed in panel A and B is slightly different since not all genes were significantly regulated along pseudotime. From there we selected the late-developmental upregulated to show their state in panel D. To clarify our approach for the readers, we added following explanation to the results section: “The number of upregulated genes in each of the two RNA-seq datasets and the determined ASIGs were presented in a SWNE plot (Figures 3-7, panel A). Genes significantly changing expression along pseudotime (Figures 3-7, panel B), were displayed in a heatmap (Figures 3-7, panel C). Finally, only the terminally upregulated genes were depicted in a tSNE plot (Figures 3-7, panel D).”. In addition, we added following paragraph to the materials and methods section: “The differential gene test was performed according to the formula “~sm.ns(Pseudotime)”, with relative expression values (relative_expr=TRUE). Subsequently, just the significant genes were subset (qval < 0.1) and plotted on the heatmap.”.

6.    Some methodological details are missing, for instance:
•    Line 108-112: not clear what was used as the control for HCC

We thank the reviewer for the opportunity to clarify this. For the HCC dataset, tumor-associated macrophages (TAM), T cells, B cells, cancer-associated fibroblasts (CAF), tumor-associated endothelial cells (TEF) and unclassified cells were sub-summarized (“other”) and compared to the malignant hepatocytes. In the text, we added: “The annotation for the HCC dataset was provided in the original study where it was referred to as “Type” [52]. The HCC dataset contained malignant hepatocytes as well as tumor-associated macrophages, T cells, B cells, cancer-associated fibroblasts, tumor-associated endothelial cells and unclassified cells. All of them (“Other”) were compared to malignant hepatocytes for the overall comparison (“Malignant cells”) ([52] GSE151530).”. In addition, we added following explanations to the manuscript: “The differential gene test was performed according to the formula “~sm.ns(Pseudotime)”, with relative expression values (relative_expr=TRUE). Subsequently, just the significant genes were subset (qval < 0.1) and plotted on the heatmap”.

  •    Nothing is mentioned about numbers of cells that were included in the scRNAseq data nor about relative contributions of the different donors.

We apologize for the lack of transparency. We added Supplementary Table 3, which indicates the individual cells per donor. In addition, we stated “The number of cells which were contributed by each donor is shown in Supplementary Table 2.” in the materials and methods section.

  •    Line 159-160: include references for the 17 studies

After initially citing the references in the materials and methods section, we now added the references to the respective section as well. We changed the sentence into “For this purpose, we performed literature research and obtained 1,535 aging/senescence-induced genes (ASIGs) from 17 studies ([12,14,29–33] and MSigDB gene sets M13158, M14701, M27188, M11850, M40058, M39619, M16568, M24705, M14683 © Broad Institute, Inc., Massachusetts Institute of Technology, and Regents of the University of California), resulting in a total number of 1,153 genes after removing duplicates (Supplementary Table S1).”.

  •    Fig. 3-7: how exactly were the 'previously defined AIGs' selected?

We apologize for the missing clarity. The genes we selected for the scRNA-seq were the genes enriched in both of the two RNA-Seq datasets as well as the ASIGs. For the sake of clarity, we added following description to the manuscript: “The number of upregulated genes in each of the two RNA-seq datasets and the determined ASIGs were presented in a SWNE plot (Figures 3-7, panel A). Genes significantly changing expression along pseudotime (Figures 3-7, panel B), were displayed in a heatmap (Figures 3-7, panel C). Finally, only the terminally upregulated genes were depicted in a tSNE plot (Figures 3-7, panel D).”.

Round 2

Reviewer 1 Report

In the revision of the original articles, the authors clarified the concept of the study (using the overexpressed genes only) and improve significantly the transparency of the gene used for the analysis. The authors also explained in more detail the heterogeneity observed in the bulk RNA sequencing by providing the age of patients in the study. Similarly, the pseudotime interpretation was clarified and the conclusion of the paper adjusted.

1) I agree with the authors that SWNE has methodological advantages on the t-SNE and that plotting the ASIGs also allows to visualize key features playing a critical role in the experiment. Could the authors provide an interpretation of the homogeneous distribution of ASIGs in the SWNE plots?

2) Similarly, could the authors interpret the very narrow clustering of ASIGs in tSNE plots and the potential biological meaning/interpretation of the cell cluster highlighted in the merged plots?

3) The authors provided a R notebook to better illustrate the exact setting of the differential gene expression analysis. However, the notebook is requiring datasets curated manually using specific format on an excel sheet. We do not think that this is a best practice for a notebook as it should run in a reproducible manner for users. Because 9 out 10 bulk RNA-seq dataset is provided from GEO, and because the notebook is for R, we would like to ask the authors to revise the notebook using a tool to retrieve the data from GEO (for example GEOquery) and generating the list of DGE without the need of manual formatting of dataset from GEO. More details on the concept could be found here: https://doi.org/10.1371/journal.pcbi.1007007 (no need to quote this paper).

4) Similarly for the single-cell RNA-seq data, as most of the datasets are from GEO, we would like to ask the authors to provide a notebook helping the readers to reproduce the figures. This will not only help the reader to understand how the tools were configured, but also increase the attractivity of the publication by helping the reader to reproduce the strategy for different cancer types or datasets. Here we strongly advise using specific tools to keep the code reproducible also in future (versioning of packages, for example described here: https://doi.org/10.1371/journal.pcbi.1008316).

Author Response

In the revision of the original articles, the authors clarified the concept of the study (using the overexpressed genes only) and improve significantly the transparency of the gene used for the analysis. The authors also explained in more detail the heterogeneity observed in the bulk RNA sequencing by providing the age of patients in the study. Similarly, the pseudotime interpretation was clarified and the conclusion of the paper adjusted.

1) I agree with the authors that SWNE has methodological advantages on the t-SNE and that plotting the ASIGs also allows to visualize key features playing a critical role in the experiment. Could the authors provide an interpretation of the homogeneous distribution of ASIGs in the SWNE plots?

We thank the reviewer for the opportunity to clarify this aspect. We were initially surprised that the chosen ASIGs were distributed homogeneously across the SWNE plot. Since we analyzed different numbers of ASIGs in the scRNA-seq data sets, we ensured to choose the same technical approach (i.e. the number of factors to iterate (k=16) and a small alpha to not move the cells too far away from the ASIGs (alpha.plot=0.4, since values>2 distort the data)). Overall, we believe that the effect is of biological rather than of technical nature. In fact, we assume that the characteristics of the ASIGs themselves are highly heterogeneous with regards to their biological function. For instance, some ASIGs encode cell cycle-controlling factors while others encode cytokine receptors or transcription factors. Due to this heterogeneity, these ASIGs play essential roles for distinct SWNE continents. Since they are functionally diverse, they may all represent important drivers but for different subpopulations. Interestingly, this corresponds to intratumoral heterogeneity. Therefore, the ASIGs appear to reflect transcriptional heterogeneity which can be biologically detected in cancer tissues. Whether this is a reason or a consequence, could be elucidated in future experimental studies.

To address this aspect, following sentence was added to the results section (exemplarily for the MM cells in Figure 3): “Interestingly, the transcriptional heterogeneity of malignant cells is reflected by the scattered distribution of the ASIGs.”. In addition, to highlight the similarity of this finding among cancer types, we added following statement to the discussion: “Interestingly, we found a heterogeneous distribution of the ASIGs in the SWNE plot of all analyzed cancer types. This finding suggests that the selected genes mirror the biological heterogeneity of malignant cell subpopulations.”.

2) Similarly, could the authors interpret the very narrow clustering of ASIGs in tSNE plots and the potential biological meaning/interpretation of the cell cluster highlighted in the merged plots?

We thank the reviewer for this suggestion and would like to clarify our approach. Pseudotime was used to narrow down the ASIGs which were significantly upregulated along a pseudotemporal development. There may be tumor-specific ASIGs which are constantly expressed over time and do not change their expression pattern once a cell has become “cancerous”. We focused on the genes which increased in expression levels over time. Interestingly, cancer cells can further develop specific aging-associated patterns as displayed in our tSNE plots. This specific cluster might behave differently compared to the other cell types and tSNE continents. To address this aspect and to improve the clarity for the readers, we added following paragraph to the discussion section: “The upregulation of specific ASIGs during pseudotemporal development was restricted to a narrow population in the tSNE plot. This finding suggests the presence of a distinct cellular population which acquired the expression of these temporal-dependent genes, even in a malignant state.”.

3) The authors provided a R notebook to better illustrate the exact setting of the differential gene expression analysis. However, the notebook is requiring datasets curated manually using specific format on an excel sheet. We do not think that this is a best practice for a notebook as it should run in a reproducible manner for users. Because 9 out 10 bulk RNA-seq dataset is provided from GEO, and because the notebook is for R, we would like to ask the authors to revise the notebook using a tool to retrieve the data from GEO (for example GEOquery) and generating the list of DGE without the need of manual formatting of dataset from GEO. More details on the concept could be found here: https://doi.org/10.1371/journal.pcbi.1007007 (no need to quote this paper).

We agree with the reviewer and are grateful for this suggestion as well as providing supporting literature. We make use of the GEOquery package regularly and highly appreciate this tool. Unfortunately, the different approaches to deposit datasets made it challenging to retrieve publicly available data. For example, in GSE119794, a *.tar file was provided, consisting of 40 *.txt.gz files. In another study, one txt.gz-file with one table containing all samples was available which was ideal for subsequently using the getGEO command. To address the reviewer’s suggestion, we redesigned the RNA-Seq notebook to ensure the reproducibility and transparency of our analyses. We now provide a R notebook which completely autonomously generates a rawcounts-file, a list of differentially expressed genes, a PCA plot as well as a simple volcano plot without the need of manual formatting. Please find the notebook “RNA_seq_PDAC.Rmd” attached to the revised version of the manuscript. Even without the necessary packages, a start from scratch is possible with this notebook in order to generate all RNA-sequencing files (exemplarily done for GSE119794). We hope that the reviewer is satisfied with the updated notebook and thank them for the opportunity to make our notebook more accessible and elegant.

4) Similarly for the single-cell RNA-seq data, as most of the datasets are from GEO, we would like to ask the authors to provide a notebook helping the readers to reproduce the figures. This will not only help the reader to understand how the tools were configured, but also increase the attractivity of the publication by helping the reader to reproduce the strategy for different cancer types or datasets. Here we strongly advise using specific tools to keep the code reproducible also in future (versioning of packages, for example described here: https://doi.org/10.1371/journal.pcbi.1008316).

We thank the reviewer for this suggestion. We now provide the notebook R Notebook “single_cell_RNA_Seq_MM.Rmd” which can be used to generate all scRNA-seq plots shown in our study (exemplarily done for Figure 3). The way we designed the notebook, all packages should be installed automatically. In this fashion, even an “empty” R package environment can be used. We advise to run the notebook on a server, since the used dataset is large (23 GB), and a local environment might not be able to handle some of the accruing vectors. Also, we would like to mention that some steps are computationally intensive, and the whole script takes several hours to run. We ensured to state all package versions within our manuscript. We hope that this notebook will help the readers of “Cells” to reproduce our figures and apply our algorithm to further datasets.

Reviewer 2 Report

The revised version of the manuscript by Saul and Kosinsky addresses most of the criticisms I raised to the previous version. I found a general improvement of the quality of the work.

Author Response

The revised version of the manuscript by Saul and Kosinsky addresses most of the criticisms I raised to the previous version. I found a general improvement of the quality of the work.

We would like to thank the reviewer for their constructive criticism and suggestions which helped to substantially improve this study.

Reviewer 3 Report

The authors' efforts to address all the comments are very much appreciated. The clarity has been improved and some missing aspects have been added to the discussion.

One small error: the new supplementary table 3 is referred to as 'Supplementary Table 2' in the Methods section. Please correct this.

Author Response

The authors' efforts to address all the comments are very much appreciated. The clarity has been improved and some missing aspects have been added to the discussion.

One small error: the new supplementary table 3 is referred to as 'Supplementary Table 2' in the Methods section. Please correct this.

We thank the reviewer for pointing out this error which was corrected in the revised manuscript. We are grateful for the reviewer’s excellent suggestions and constructive criticism which substantially improved this study.

Round 3

Reviewer 1 Report

The manuscript was corrected accordingly to my previous comment and gain in reproducibility.

Reviewer 2 Report

Only minor corrections have been made to the manuscript. As for the previous version, I find that Authors have substantially improved the original manuscript.

Reviewer 3 Report

Congratulations with this work; I have no further comments.